# Next-generation seismic model of the Australian crust from synchronous and asynchronous ambient noise imaging

Yunfeng Chen [1,2] ✉, Erdinc Saygin [2], Brian Kennett [3], Mehdi Tork Qashqai [2], Juerg Hauser [2], David Lumley [4] & Mike Sandiford [5]

The proliferation of seismic networks in Australia has laid the groundwork for high-resolution probing of the continental crust. Here we develop an updated 3D shear-velocity model using a large dataset containing nearly 30 years of seismic recordings from over 1600 stations. A recently-developed ambient noise imaging workflow enables improved data analysis by integrating asynchronous arrays across the continent. This model reveals fine-scale crustal structures at a lateral resolution of approximately 1-degree in most parts of the continent, highlighted by 1) shallow low velocities (<3.2 km/s) well correlated with the locations of known sedimentary basins, 2) consistently faster velocities beneath discovered mineral deposits, suggesting a whole-crustal control on the mineral deposition process, and 3) distinctive crustal layering and improved characterization of depth and sharpness of the crust-mantle transition. Our model sheds light on undercover mineral exploration and inspires future multi-disciplinary studies for a more comprehensive understanding of the mineral systems in Australia.

The Australian continent consists of a collage of crustal domains that have been successively accreted to the Archean cores during three supercontinent cycles[1]. With approximately 80 percent of its landmass covered by extensive Post-Mesozoic sediments and regolith[2], the basement of the Australian continent is composed of metamorphic rocks with a broad age spectrum of over 4 billion years, ranging from the Archean and Proterozoic cratons in west and center of Australia to the Phanerozoic accreted terranes in the east (Fig. 1a). Understanding of Australian lithospheric structure has been significantly advanced in the past decades, thanks to extensive active and passive seismic surveys conducted across the continent. The synthesis of seismological constraints (see refs. [2,3] for a review) has led to the development of the community reference models of the Moho depth (AusMoho[4]) and crustal and mantle elastic properties (AuSREM[5,6]). Among these pioneering works, ambient noise imaging has played a major role in

mapping shear velocity structures at a continental scale[7–9]. Following the large-scale investigations, ambient noise studies took advantage of dense temporary seismic deployments and continued refining the crustal structures beneath the footprints of the portable arrays[10–19]. One excellent example is the WOMBAT transportable array, which achieved a station spacing of 50 km and significantly improved the data coverage in a large portion of eastern Australia[13]. The outcomes based on these high-quality array data offered critical constraints to regional crustal velocity[12,14,20] and anisotropy structures[15,21].

Substantial progress in resolving crustal structure has been made through these earlier studies. However, challenges remain in further improving the resolution and depth sensitivity of the current models. The most recent continental-scale shear velocity model was developed almost a decade ago based on sparsely distributed seismic stations[8]. While this model represents an excellent first-order estimate of shear

[1]Key Laboratory of Geoscience Big Data and Deep Resource of Zhejiang Province, School of Earth Sciences, Zhejiang University, Hangzhou 310027, China. [2]Deep Earth Imaging, Future Science Platform, CSIRO, Kensington, WA 6151, Australia. [3]Research School of Earth Sciences, Australian National University, Canberra, ACT 2601, Australia. [4]Departments of Geosciences, Physics, University of Texas at Dallas, Richardson, TX, USA. [5]School of Earth Sciences, University of Melbourne, 3010 Melbourne, Australia. ✉e-mail: yunfeng_chen@zju.edu.cn

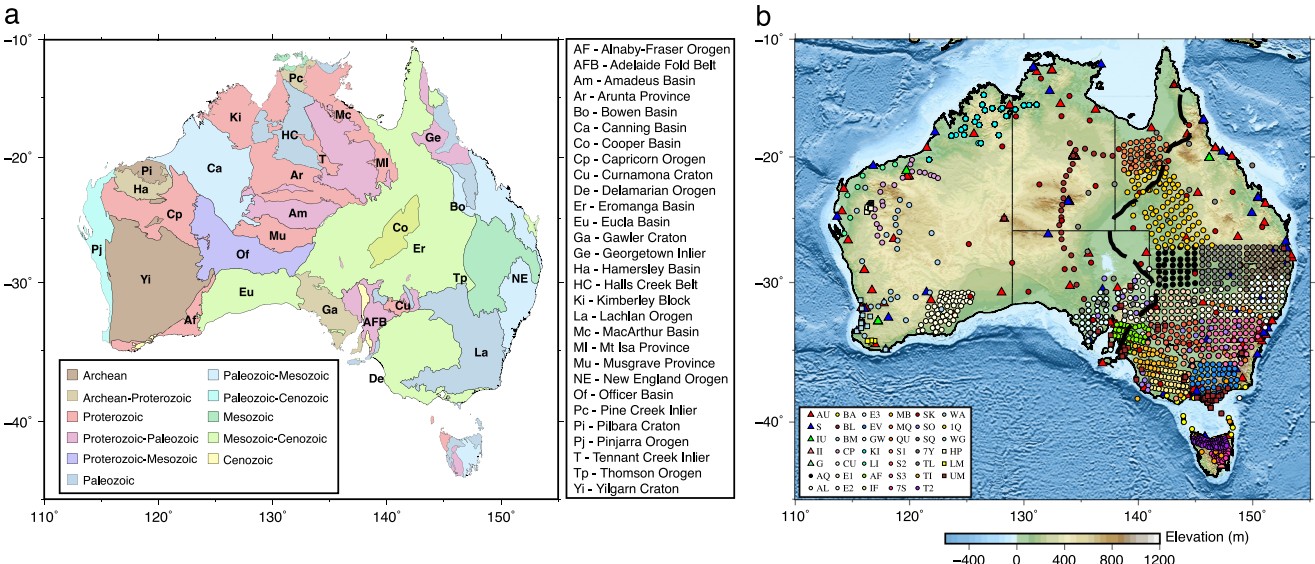

**Fig. 1 | Tectonic subdivision and station distribution of the Australian continent. a** Major geological domains. The domain boundaries and the outline of the Cooper basin are from ref. [47]. **b** Distribution of seismic stations used in this study, inlcuding historical deployments through to 2019. The background is topography from ETOPO1. The black dashed indicates the location of the proposed Tasman line from ref. [25].

velocity structures, its resolution is far from ideal as evidenced by (1) an overall low horizontal resolution (about 300 km) at even the best-constrained depth range in the upper crust, (2) poorly resolved middle to lower crustal structures, particularly at depths below 30 km, and (3) minimal sensitivity to lower crust and the Moho. These resolution issues are partially caused by inversion with a fixed grid (i.e, non-data adaptive) parameterization[8] but, more importantly, are consequences of limited and uneven spatiotemporal sampling of seismic data. Despite a rapid expansion of seismic networks, the spatial distribution of sensors is still biased towards the south-east of Australia, whereas the western portion (e.g., the desert areas of central Australia) is only sparsely sampled due to logistical challenges (Fig. 1b). On the other hand, the temporal distribution of data is highly irregular, centering on relatively short durations of typically less than two years when a portable array is operating (supplementary Fig. S1). Continuous long-duration recordings are only available from ~200 permanent stations that form the backbone of the national seismograph network. Consequently, challenges remain to reconcile different spatio-temporal characteristics (e.g., operating period, array coverage, and resolving power) between the permanent and temporary networks.

Developments in both seismic instrumentation and imaging techniques have facilitated a new appraisal of the Australian crust. Here, we tackle the data related issues by (1) taking advantage of the long recording time from permanent stations and dense spatial coverage offered by portable arrays, and, equally importantly, (2) developing a higher-order ambient noise imaging workflow based on a recently proposed technique of cross-correlation of ambient noise correlation functions ($C^2$)[22,23]. This new workflow allows us to reconstruct the noise correlation functions between a pair of asynchronous stations (e.g., stations from two portable arrays deployed at different times) via surrounding long-operating stations, a task that is not feasible with the conventional ambient noise correlation approach (i.e., $C^1$). The combination of an extensive dataset and the innovative $C^2$ technique enables us to exploit the resolving power of the ambient noise data and develop a new shear velocity model of the Australian crust at much improved resolution. Our model contributes to an improved understanding of the Australian crust at both shallow (e.g., sedimentary distribution and cover thickness) and deeper (e.g., crust-mantle transition) depths. The detailed basin structures and deep crustal architectures provide useful guidance for undercover mineral exploration. On the other hand, the new knowledge of crustal interfaces and crust-mantle transition enables constructing a continental-scale model of structural layering for the first time. These integrated structural constraints pave the way for developing the next-generation seismological reference model of the Australian continent.

## Results

### Improvement of ray path coverage

Seismic ray path coverage is significantly improved upon previous studies owing to the rapid growth of seismic stations and the development of the $C^2$ workflow (see "Methods" section). This technical improvement enables us to incorporate asynchronous stations into data analysis, which otherwise cannot be utilized for ambient noise imaging (supplementary Figs. S2 and S3). At short periods, the ray paths from $C^1$ mostly sample the south and southeast of Australia. In comparison, the ray path density decreases northward except for a high-density band in central Australia following the BILBY seismic array (network code BL in Fig. 1b). Major data gaps exist in northwestern and central-eastern parts of the continent (Fig. 2a). The $C^2$ approach effectively bridges the asynchronous networks deployed in the west and southeast of Australia, leading to a much higher ray-path density in these regions (Fig. 2b). The number of ray paths also increases significantly along the southern margin of the Australian continent by bridging the temporary network (ALFREX, network code AL in Fig. 1b) deployed to the east of the Yilgarn craton with the portable arrays in southeastern Australia. The number of combined ray paths from $C^1$ and $C^2$ is twice that available from $C^1$ alone (Fig. 2c), resulting in a denser and more homogeneous data distribution. A similar improvement is made at long periods, leading to a more balanced ray-path coverage in the combined dataset (Fig. 2d, e, f).

### Group velocity structure

The group velocity travel times between 4 and 46 seconds are measured with FTAN[24] and maps of lateral velocity variation are obtained by inverting the Rayleigh wave travel times with trans-dimensional Bayesian inversion[9] (see Methods section). The inversion implements an adaptive parameterization with Voronoi cells that considers the varying spatial density of the seismic data (see Fig. 2). The resulting group velocity maps at short periods (e.g., 6 s and 14 s) show an elongated band of prominent low velocities from northwest to east of

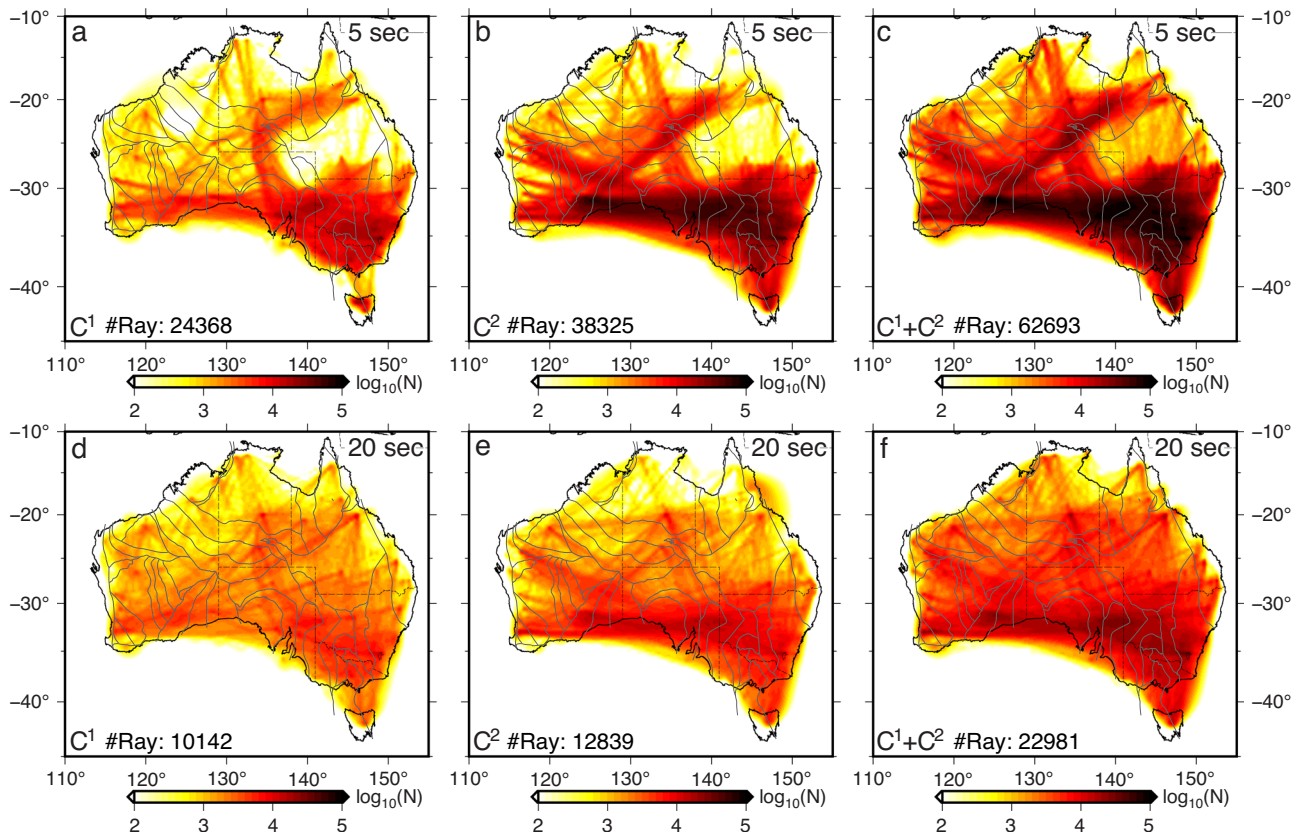

**Fig. 2 | Comparison of the ray-path density retrieved using different methods.** Ray-path density at two periods from (**a**, **d**) $C^1$, (**b**, **e**) $C^2$ and (**c**, **f**) combined $C^1$ and $C^2$ functions. The total number of ray paths is labeled. The gray lines indicate the major tectonic boundaries[48].

Australia, coinciding with the distribution of large-scale sedimentary basins (Fig. 3a, b). Small-scale low-velocity (<2.8 km/s) anomalies are observed in the offshore areas within the Bass Strait and persist to intermediate periods of 22 s (Fig. 3c). At longer periods, below-average velocities (e.g., <3.4 km/s at 30 s period) are observed in central Australia (Fig. 3d, e), which form a low-velocity structure extending in NE-SW directions. The strength of the low-velocity anomalies diminishes at the longest period of our observations (Fig. 3f). On the other hand, persistent high velocities that are at least 5% greater than the regional average dominate western Australia at almost all periods, whereas the north and south of Australia are mainly characterized by high velocities at short (<14 s) periods (see Fig. 3a).

**Shear velocity structure**

We invert group velocities to estimate 1D shear velocity profiles on a grid of nodes at 0.75-deg spacing with a linear inversion scheme (see "Methods" section). At shallow depths, a prominent low-velocity channel of less than 3.0 km/s underlies the northwest of Australia and transitions into a broad low-velocity (<3.4 km/s) zone in the east (Fig. 4a, b). High velocities above 3.5 km/s are observed in the cratonic regions of western, southern, and northern Australia. A prominent streak of high-velocity (>3.6 km/s) structure underlines western Australia, trending NE-SW, and extends to ~8 km depth. At the shallow depth of 1 km, this high-velocity structure bifurcates into a relatively weaker (<3.6 km/s) branch that extends southward to the continental margin. At middle crustal depths between 10 and 25 km, seismic velocity in the eastern half of the Australian continent is on average 0.04 km/s lower than in its western counterpart (Fig. 4c, d), though the transition between these two regimes is not well defined. Crust with below-average shear velocity gradually expands towards the west at greater depths and terminates sharply at the eastern edge of the Yilgarn craton (Fig. 4e). At the bottom

of the crust, a band of low-velocity (~4.0 km/s) zone resides in the center of Australia and is elongated in NE-SW direction, forming a sharp velocity contrast of about 5% with the surrounding high-velocity (>4.4 km/s) regions (Fig. 4f). The boundary between the low- and high-velocity crust in eastern Australia is located near the various versions of the Tasman line[25].

Cross-sectional views of our model show significant velocity variation at shallow depths (Fig. 5), where a large portion of the profiles are underlain by low-velocity (<3.4 km/s) structures. The distribution of these low-velocity regions agrees well with the location of known sedimentary basins in both onshore and offshore areas. For example, a narrow (~300 km) and deep (~8 km) low-velocity zone is observed beneath the Bass Strait (Fig. 5g; also see Fig. 4a). The middle crust is relatively homogeneous compared with the highly heterogeneous upper crust, with slightly higher velocities (3.8–3.9 km/s) observed beneath the Canning and Officer basins (Fig. 5a–d). Lower crustal velocities generally vary between 3.9 and 4.3 km/s while showing significant perturbations in the middle portion of the E-W trending profiles where the topography is relatively high (see Fig. 5a–c). The Moho is determined using a velocity gradient approach (see "Method" section) and shows a first-order agreement with the Moho depths defined in the AusMoho model[4], but also reveals new details of the Moho variation across the continent.

## Discussion

The current crustal component of the Australian seismological reference model (AuSREM)[5] is primarily constructed from active seismic refraction and wide-angle reflection data, complemented by information from passive seismic surveys based on receiver function, teleseismic tomography and ambient noise imaging. We compare the average velocities between our model and AuSREM at upper, middle

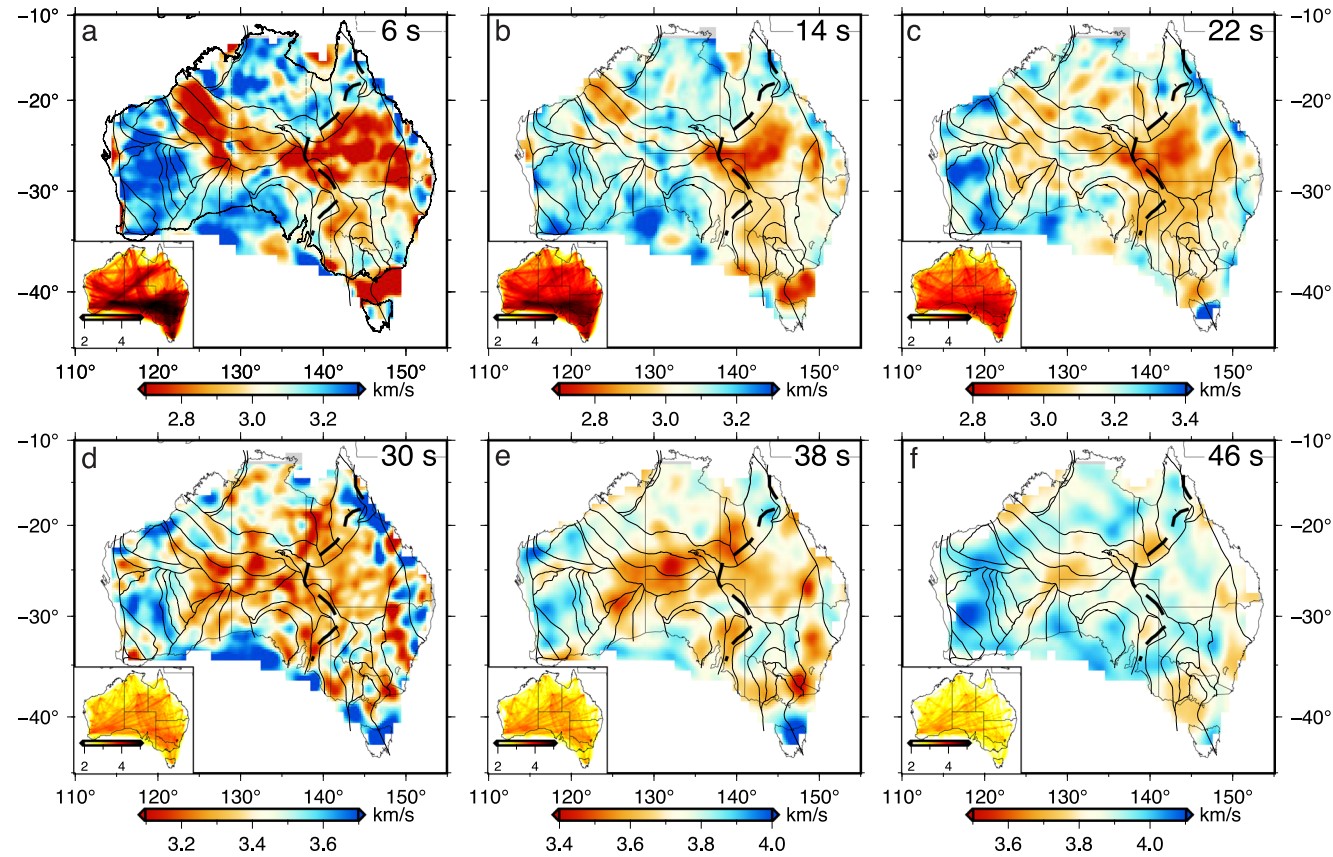

**Fig. 3 | Surface wave group velocities.** Group velocity maps at periods of **a** 6 s, **b** 14 s, **c** 22 s, **d** 30 s, **e** 38 s, and **f** 46 s. The Archean cratonic regions are labeled in **a**. The black lines indicate the major tectonic boundaries[48]. The inset shows the ray-path coverage in logarithmic scale at the corresponding periods. Ga Gawler Craton, Pi Pilbara Craton, Yi Yilgarn Craton.

and lower crustal depths (Fig. 6). While the upper crust is dominated by low velocities in sedimentary basins in both models (Fig. 6a, d), the shallow structures are particularly well constrained by our large dataset as evidenced by 1) a better spatial correlation between the distributions of large-scale velocity structures and tectonic domains, 2) an improved clarity of the geometry of low-velocity depocenters (e.g., Canning and Cooper basins), and 3) clearly resolved small-scale anomalies of, for example, the Bowen Basin in eastern Australia. The major differences are observed in eastern Australia where the velocity in our model is on average 0.1 km/s lower than that of AuSREM, owing to the presence of deep and broad low-velocity structures beneath major sedimentary basins (Fig. 6g). In the middle crust, the lateral velocity variation is relatively small with a standard deviation of 0.067 km/s (Fig. 6b), which is about half of that of the upper crust (0.12 km/s), suggesting a more homogeneous middle crust. Both models show scattered anomalies with an overall eastward decreasing trend of seismic velocity (Fig. 6b, e). The velocity difference between the two models exhibits a less clear pattern compared to the upper crust, with a mean value of 0.08 km/s and a standard deviation of 0.06 km/s (Fig. 6h). The lower crust shares a similar first-order structural variation in the two models including the dominant low velocities in central Australia as well as local-scale anomalies such as a prominent high-velocity zone beneath Tasmania (Fig. 6c, f). However, the values of velocity anomalies in AuSREM are generally higher (by ~0.1 km/s) than those defined in our model (Fig. 6i). We speculate that this is caused by a lack of direct shear velocity constraints for the lower crust, particularly at depths below 30 km, in AuSREM. Specifically, long-period data was not available in earlier ambient noise studies and hence the shear velocities of AuSREM are scaled from the P-wave velocities using the Vp/Vs ratio derived from receiver function

inversions. The uncertainties in both parameters may lead to spurious structures that follow the acquisition footprint. For example, AuSREM shows an intriguing NE-SW trending structure with a below-average velocity that extends across western Australia (near 30°S, 120°E), whereas our model is defined by high velocities that correlate well with the distribution of the Yilgarn craton.

The combined $C^1$ and $C^2$ dataset enables us to resolve shallow crustal structures at much-improved resolution. A vast area of the Australian continent is dominated by low velocities at shallow depths (Fig. 7a), which we attribute to thick sedimentary strata and highly weathered rocks (i.e., the regolith) that cover about 80% of the Australian landmass. The distribution of seismic velocities shows a distinctive relationship with the locations of operating mines and known mineral deposits (see Fig. 7a). A significant portion of operating mines are located in high-velocity regions that mark the exposed shield including, most notably, the Yilgarn craton in Western Australia that is a major mineral province in Australia. In this region, the majority of mineral deposits are located along several NW-SE trending domain boundaries in the eastern half of the Yilgarn craton. However, deviating from this regional trend is a NE-SW oriented mineralization zone that transects the central portion of the craton. The distribution of these mineral deposits is well correlated with a band of high-velocity (>3.6 km/s) structure, which may indicate an unexposed structural lineament that controls the mineral deposition process. On the other hand, most of the mineral deposits in central-eastern Australia are found near the edges of the low-velocity zones associated with shallow sediments (e.g., Darling and Surat basins). This observation could reflect either (1) a preferential generation and concentration of mineral deposits, and/or (2) the challenges in exploring minerals underneath the cover.

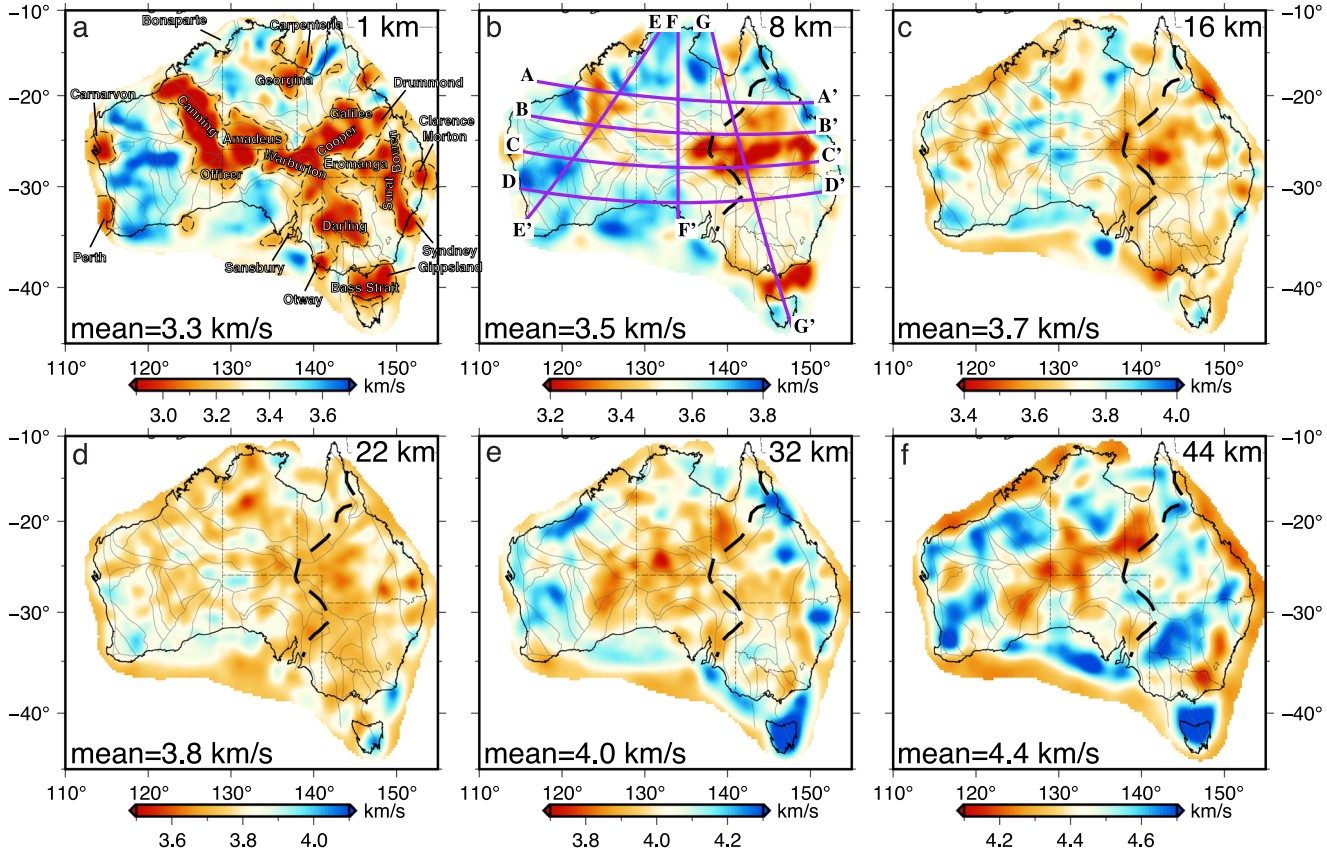

**Fig. 4 | Shear velocities at six depth ranges.** The black dashed lines in **a** indicate the velocity contour of 3.3 km/s. Major sedimentary basins are labeled. The location of the proposed Tasman line from ref. [25] is marked by the black dashed line in **b**–**f**. The gray solid lines indicate the major tectonic boundaries[48] and the purple lines in **b** mark the locations of cross-sections shown in Fig. 5.

To assess the former hypothesis, we conduct a quantitative analysis to examine the spatial correlation between seismic velocity and mineral deposit location (see "Method" section for details). Our result shows that only about 20% of mineral deposits are located in regions characterized by relatively low velocities (<3.3 km/s) in the shallow 5 km, whereas this ratio increases to 45% while assuming a randomly distributed mineral locations (Fig. 7b). We further extract velocities in different depth intervals to examine if such a relationship persists to greater depths. To reduce the sampling bias caused by the clustering of deposits, the nearby deposits (within a 0.2-deg cell) are grouped to form a single sampling point. Our analysis shows that mean shear velocities are consistently faster beneath the mineral deposits than the model average (Fig. 8). The mean velocities of the mineral deposit group are 0.06 km/s faster in the shallow (0–10 km) crust and 0.03 km/s faster in the deep (30–40 km) crust and are slightly faster (~0.01 km/s) in the middle crust (10–30 km). The reliability of the difference in mean values of the two distributions is evaluated using the $t$-test, which assesses the validity of the hypothesis that the mean of the mineral deposit group is greater than the continental average. We obtain large $t$ scores at all depths including the middle crust where the velocity difference is small, showing a $t$ score of 2.94 with a $p$-value of 0.002 (see Fig. 8). The test results indicate that the observation of consistently faster crustal velocities beneath the mineral deposits is statistically significant. This distinctive pattern suggests that the mineralization process likely involves the whole crust, not just the shallow portion. A corollary is that the distribution of mineral deposits may not solely reflect the sampling bias in mineral exploration due to the presence of sedimentary cover (e.g., the location where outcrop exists). There has been growing evidence that the formation of mineral deposits is closely related to deep magmatic processes controlled by lithospheric-scale structures[26–28]. For instance, a recent study has reported a close spatial association between lithospheric gradient zones and sediment-hosted deposits around the globe[27]. In Australia, this study showed that giant mines were preferentially located within 100 km of the craton edge which marks a transition in lithospheric thickness[27]. One possible mechanism to form such a lithospheric boundary is through a continental rifting event in an extensional setting[29,30], during which a basin subsides as a consequence of syn-rift mechanical stretching and post-rift isostatic re-equilibration[30]. However, not all basins and their associated sediment-hosted deposits are related to continental rifting on the craton margin. A notable exception is the Bowen basin in eastern Australia which has undergone contemporaneous thermal and foreland-loading subsidence in the Late Permian[31]. Nonetheless, the deep-seated structures such as the basin-bounding faults could facilitate the transport of geothermal fluids and thus play a critical role in the genesis and concentration of the sediment-hosted base metal deposits near the basin margins. Away from the basin margins, there is a significant portion of mineral deposits located in high-velocity areas of the cratonic region in Western Australia. These deposits yet still form prominent clusters that align parallel to domain boundaries or elongate E-W along a high-velocity structure in central Yilgarn craton (see Fig. 7a). We argue that these structural lineaments could mark weak/fracture zones that channel the mineralizing fluids and control the deposition sites. Overall, our analysis suggests a potential whole-crustal control of the mineral distribution pattern, however, the exact mechanisms for various mineral types are likely variable across the continent. More detailed seismic imaging with dense arrays and improved knowledge of regional crustal structures are still in need to

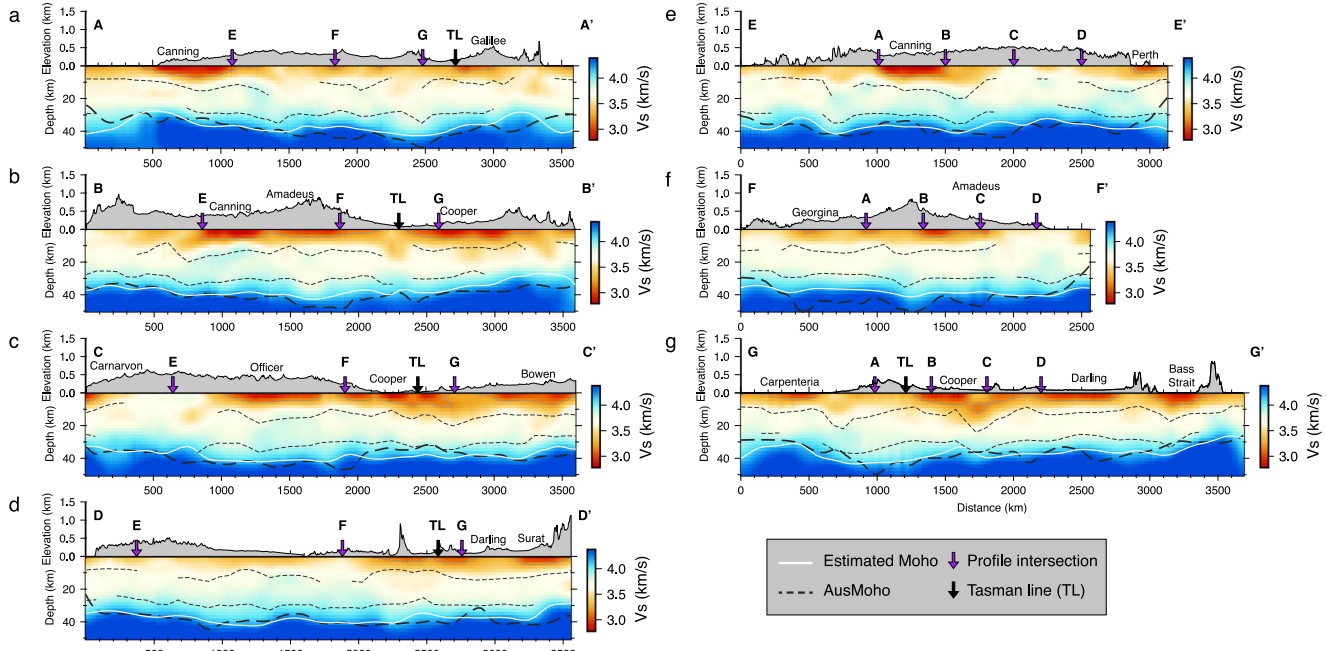

**Fig. 5 | Cross-sections showing lateral and vertical variation of shear velocities. a–d** Four east-west oriented profiles. **e** A northeast-southwest oriented profile sampling the western half of the continent. **f** A north-south oriented profile sampling central Australia. **g** A northwest-southeast oriented profile sampling the eastern half of the continent. The locations are shown in Fig. 4b. The gray solid lines indicate the velocity contour of 3.3 km/s. The gray dashed lines represent major crustal interfaces determined from velocity gradients. The Moho extracted from our model is indicated by the white line and the Moho defined by the AusMoho model is marked by the black dashed line. TL Tasman line.

quantify the contribution of different depth levels (e.g., upper vs. lower crust) to the mineral deposition process.

Unlocking the mineral potential under the deep cover lies in improving the knowledge of sedimentary basins. Ambient noise imaging allows a quantitative assessment of the spatial distribution and thickness variation of the sediments across the continent. We extract the basement depth from all one-dimensional (1D) velocity profiles using contour values between 3.1 and 3.3 km/s. We adopt a range of velocities instead of a fixed value to account for the lateral variation in average crustal velocity. This approach also accounts for the uncertainties in the group and shear velocities inversions that are difficult to quantify without a full error propagation analysis. Our synthetic analysis shows that the shallow basement depths are well constrained by short-period dispersion data (supplementary Fig. S11). The resulting sedimentary thickness map mimics the distribution of seismic velocities at shallow depths (Fig. 7c), which shows deep (> 5 km) sedimentary deposits in the Canning and Cooper basins, and offshore areas of the Bass Strait. Most of eastern Australia is covered by shallow sediments of less than 2 km (e.g., Eromanga and Darling basins) with a major depocenter observed near the Cooper basin. Medium-scale basins such as the Carnarvon and the Perth basins along the western coast are also resolved by our data. The sedimentary structures are relatively well constrained by earlier investigations, which allows us to benchmark the shallow structures in our model. We compare the obtained sedimentary thickness with that from the recently released OZ Seebase model[32] (Fig. 7d) compiled from extensive geophysical (primarily magnetic and gravity) and geological data. While these two models are derived from completely independent data sets, the resulting sedimentary structures are highly similar, with both showing significant variations in sedimentary thickness across the continent. Although the incorporation of geophysical constraints using potential field data provides a continuous mapping of the basement structures in the OZ Seebase, its model accuracy is the highest in regions where seismic and well/drillhole constraints are available[33]. Our results enable

further calibration to improve the current understanding of the sedimentary structure and highlight the potential of ambient noise imaging in undercover explorations.

Our model provides new constraints on the 3D structural variation of the crust. We determine for the first-time major crustal layering of the Australian continent using a velocity gradient approach (see "Method" section). The shallow crustal interface that approximately divides the upper and middle crust is on average 10 km deep, and is generally deeper beneath the sedimentary basins and shallower in the cratonic regions in western, northern and southern Australia (Fig. 9a). The lower interface separating the middle and lower crusts resides at ~27 km depth, which exhibits a similar pattern as the shallower one with the most significant depression observed in central Australia (Fig. 9b). Compared with a relatively well-constrained upper crust, knowledge of the lower crust, particularly its shear velocities, remains limited and thinly explored by ambient noise data in earlier studies. We extend the period of dispersion analysis up to 46 s (see Fig. 3f), which is a major improvement upon ref. [8] who conducted ambient noise imaging up to 32 s. However, the lack of existing lower crustal shear velocity information prohibits us from a direct comparison of velocity structures. Instead, we take advantage of the well-constrained Moho depths from the previous active and passive source imaging and assess our model accuracy at lower crustal depths. We extract the Moho depth following the approach proposed in ref. [34], which determines the Moho depth using a certain fraction of velocity jump from the lower to the upper mantle (see "Method" section). We find that a 50% velocity jump provides a good estimate of Moho depth compared to existing seismic constraints[4]. The resulting Moho map shows a predominantly thick (over 45 km) crust in central Australia (Fig. 9c). The crust thins towards the edges of the continent with patches of shallow Moho observed in the center of Yilgarn and Pilbara cratons in Western Australia. Two recent Moho maps are compared with our results including the recent update of AusMoho[35] (Fig. 9d) and the model from ref. [36] (Fig. 9e). The AusMoho models were compiled from a variety of data

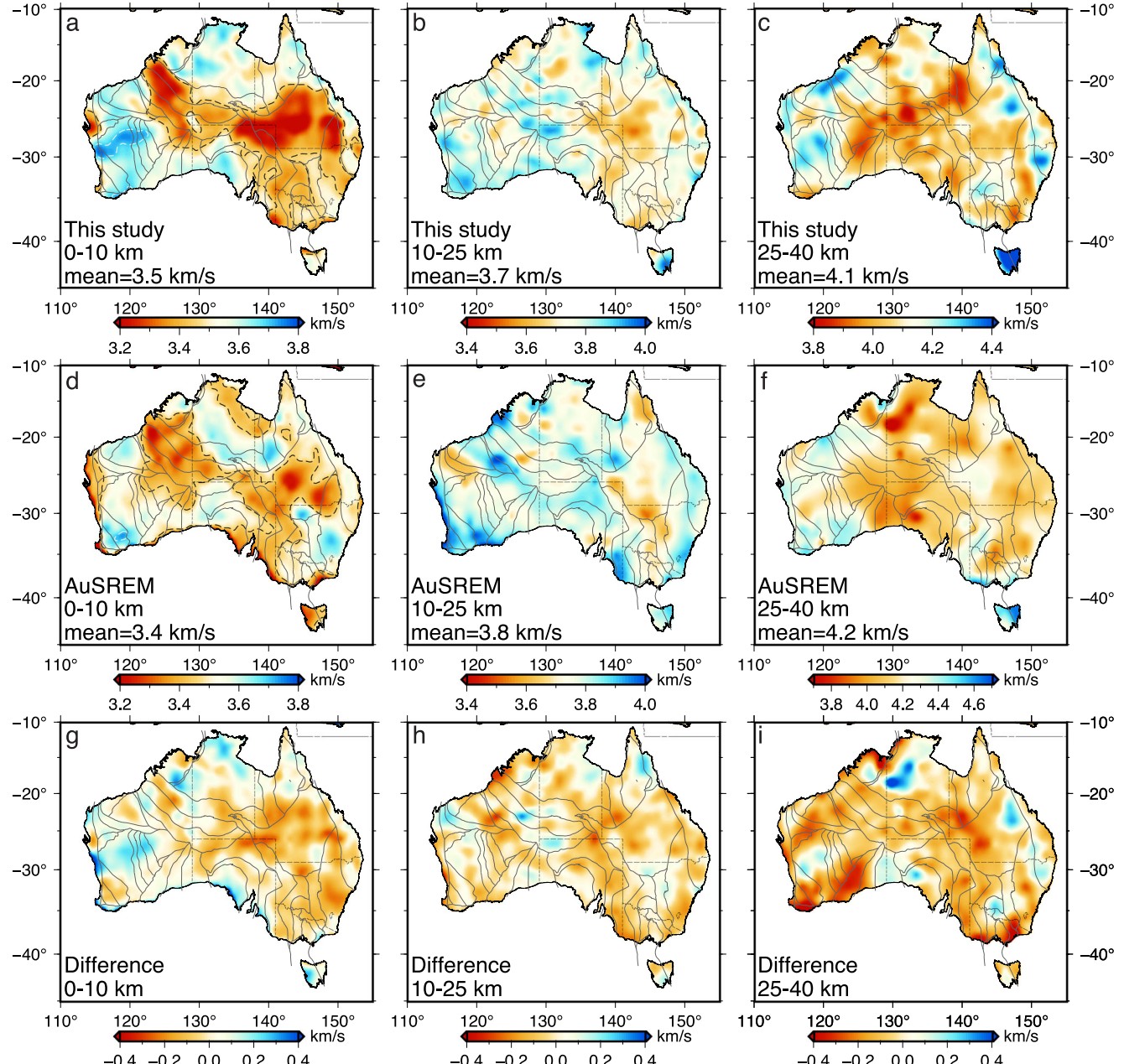

**Fig. 6 | Comparison of crustal structures between our model and AuSREM[5].** Average shear velocities of the upper, middle, and lower crust in **a**–**c** our model and **d**–**f** AuSREM. **g**–**i** The corresponding difference between the two models. The black dashed lines in **a** and **d** indicate velocity contours of 3.45 km/s.

including seismic reflection/refraction, receiver function, seismic tomography and earthquake auto-correlation[4], whereas the latter model was solely derived from auto-correlation imaging in conjunction with probabilistic coda inversion[36]. Despite differences in station distribution and data type, these models show a similar trend of Moho variation as observed in our model, with the deepest Moho underpinning central Australia and a gradual taper towards the eastern and western continental margins. The variation among these models can be attributed to the difference in data type, imaging technique and distribution of seismic stations. A significant difference among these models is centered on the northeastern edge of the Gawler craton, where the Moho is significantly shallower (by 10 km) compared with the surrounding crust. Instead, our model favors a deeper Moho in this region. In the former models, Moho is derived primarily from the point-based measurements, hence the resolution is the highest in the neighborhood of the sampling points (e.g., near active source lines and

temporary/permanent stations), yet the station coverage is particularly sparse in this region, leaving a large data gap that coincides with the region of shallow Moho. It is worth noting that the updated Aus-Moho model, which assimilates the new Moho depth estimates from a dense linear array in this region[37], shows an eastward extension of the thick crust (see the circled area in Fig. 9b) and partially improves the regional Moho constraint. Compared with receiver-based imaging methods, surface waves sample the structures along the propagation ray path. Hence, the sensitivity is not only restricted to structures beneath stations but can also provide necessary constraints to the Moho depth in the inter-station areas. Our study shows that the ambient noise data is capable of mapping large-scale Moho variations and complementing the existing Moho information in regions where in situ sampling is limited. Moho differences also exist in northern Australia where the path coverage for ambient noise imaging is sparse. Recent deployments of instrumentation associated with Geoscience

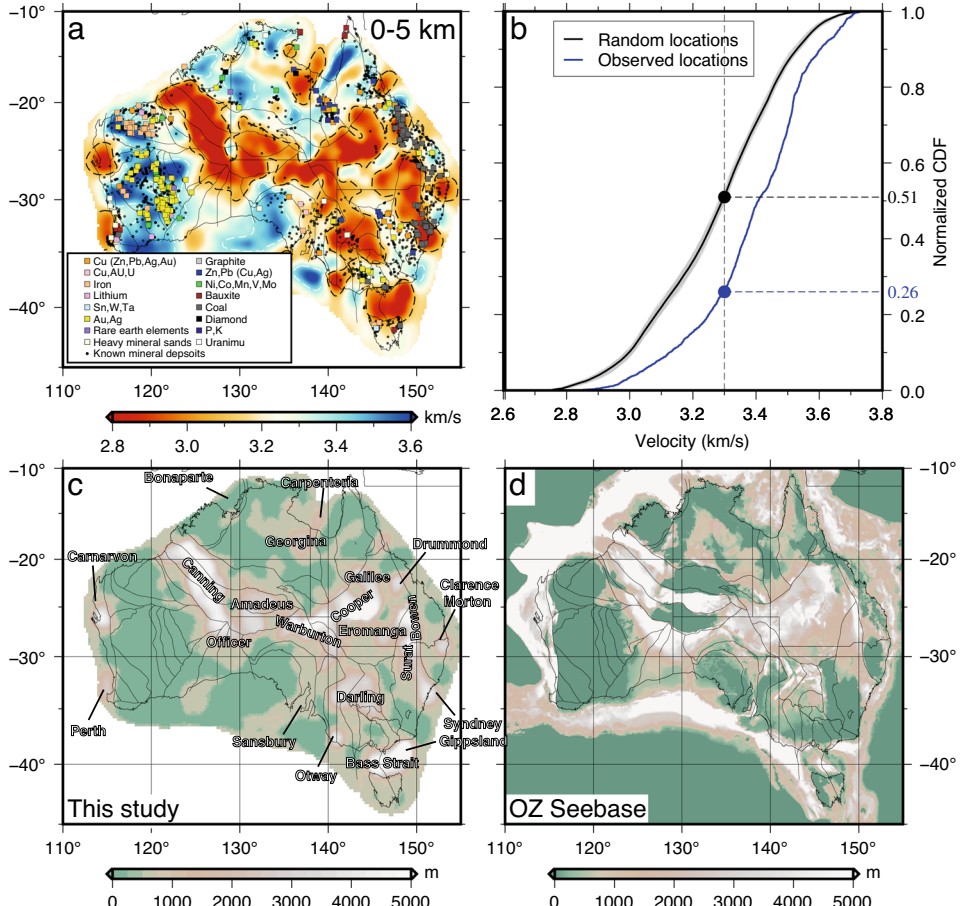

**Fig. 7 | Sedimentary basin structure. a** Average shear velocities between 0 and 5 km depths. The squares indicate the locations of operating mines[49]. The black circles indicate the locations of known mineral deposits. The white and black dashed lines highlight high and low velocity contours of 3.6 km/s and 3.3 km/s, respectively. **b** Normalized cumulative density function (CDF) of seismic velocities sampled at locations of mineral deposits in the depth range of 0–5 km. The blue line indicates the CDF calculated according to the observed mineral locations. The black line shows the mean CDF of 300 sets of randomly distributed mineral locations and the range of all trials is colored in gray. The circles indicate the intersections of CDF curves with a constant velocity of 3.3 km/s which represents the average velocity in this depth range. **c** Sedimentary thickness estimated from our model with major sedimentary basins labeled. **d** Sedimentary thickness map from the OZ Seebase model[32].

Australia's Exploring for the Future Project[38] and regional experiments such as the WA-Array will eventually help to improve coverage.

The sensitivity of dispersion data to deep structure (supplementary Figs. S10 and S12) enables us to characterize the transition in physical properties from crust to mantle. We measure the transition thickness (i.e., Moho sharpness) and its corresponding velocity variation by considering the depth and velocity difference between the 50% and 85% of velocity increases from crust to mantle[34]. The Moho sharpness (Fig. 9f) is generally anti-correlated with the velocity jump (Fig. 9g), wherein a smaller velocity increase leads to a sharper boundary and vice versa. The Moho sharpness map shows large (>6 km) transition thicknesses in the western, northern and eastern Australia, whereas zones of relatively thin (<4 km) crust-mantle transition dominate central Australia and extend southward to the coastline (see Fig. 9f). Our measurements from ambient noise imaging are compared with the constraints from receiver functions that classify the crust-mantle transition into four distinctive groups[39]. Receiver function imaging reveals large variations in transition thickness across the continent and at a regional scale of hundreds of kilometers whereas our results mostly show smoothly varying structures. The difference could reflect the different resolving power and lateral sensitivity of the two methods. Similar observations between the two studies include (1) sharp (2–4 km) Moho along the northern and southeastern edges of the Yilgarn craton and considerable variability in the cratonic interior

of Western Australia, (2) sharp Moho in southern Australia, particularly in the vicinity of the Gawler craton, and (3) thick transition regions beneath the sedimentary basins (e.g., Eramanga and Cooper basins) in eastern Australia. Overall, our observations do not show a clear relationship of Moho sharpness to tectonic age. For example, a thick crust-mantle transition is observed beneath both Archean and Phanerozoic basements. This could suggest that the rheological properties near the base of the crust are not only inherited from crustal formation but may have undergone substantial reworking during the secular evolution of the continental crust[39].

Seismic images from different depths collectively form an updated appraisal of the Australian continent (Fig. 10). The improved structural constraints from the sedimentary basins to the Moho enable us to quantify the characteristics of the sedimentary cover, map the seismic properties of its underlying basement rocks, and resolve the major crustal layering of the continent. This updated knowledge of crustal structures from a seismological perspective marks an important step forward toward understanding the formation and deposition processes of mineral resources. Our study establishes a general connection between crustal architectures and mineral deposits. However, more in-depth knowledge of the formation mechanisms of various types of mineral systems requires advance in the following aspects. First, seismic studies need to further refine the imaging resolution to regional/local-scale crustal structures and better constrain the deeper

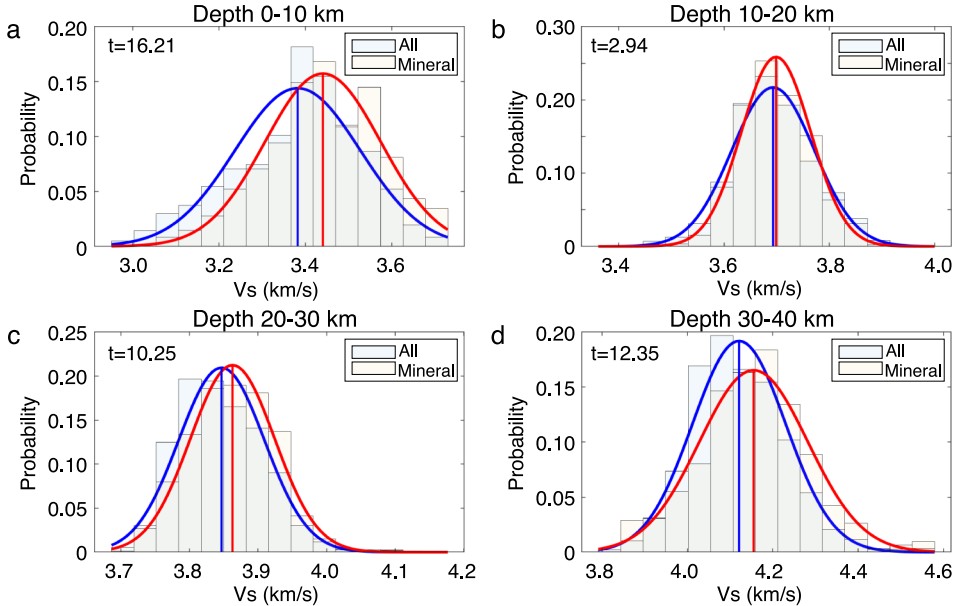

**Fig. 8 | Comparison of velocities beneath the mineral deposits and other locations.** Seismic velocities are analyzed in the depth ranges of **a** 0–10 km, **b** 10–20 km, **c** 20–30 km, and **d** 30–40 km. The histogram shows the distribution of shear velocity and is normalized. The curves in corresponding colors are the Gaussian functions that best fit the histograms. The blue and red vertical lines indicate the mean velocity for each group. The statistic value from the *t*-test is indicated in the upper left corner.

(lower crust and upper mantle) portion of the model to resolve the lithospheric-scale structures that govern the fluid flow pattern. In this regard, our new 3D shear velocity model builds a solid basis for future investigations by assimilating various data types and exploration approaches, which brings a new opportunity to develop a new seismological reference model of the Australian crust. Additionally, multi-disciplinary studies combing mineralogy, geochemistry, geology, and geophysics are required to understand the geological processes, tectonic settings, and geodynamic environments and develop a comprehensive model for mineralizing systems[40]. Finally, the new ambient noise imaging workflow developed in this study can also inspire similar continental-scale investigations for an improved understanding of the Earth's subsurface.

## Methods
### $C^1$ and $C^2$ calculations
The seismic instrumentation in Australia has increased gradually over the years from tens of stations in the early 1990s to ~300 active stations in recent years (supplementary Fig. S1a). Among these stations, several long operating networks (e.g., AU and S1) form the backbone array, which are augmented by temporary deployments with an operation period of 1–2 years in target regions across the continent (Figure S1b; also see Fig. 1b). The $C^2$ workflow is well suited for networks in Australia where permanent stations are distributed near the coastal areas, surrounding the temporary deployments further inland (see Fig. 1b). We briefly summarize the key processing steps employed to extract the noise correlation functions (NCFs) between synchronous and asynchronous stations. As a first step, conventional ambient noise correlation (i.e., $C^1$) is conducted between synchronous station pairs. The continuous seismic recordings are cut into 1-hour segments with a 30-min overlap between consecutive time windows. After removing the mean and linear trends, we apply a low-pass filter with a corner frequency of 1.25 Hz and downsample the data to 2.5 Hz. The amplitude spectra of traces are normalized (i.e., spectral whitening) to broaden the frequency content. A daily NCF is obtained by cross-correlating the preprocessed segments and stacking the resulting cross-correlation functions from all (48) time windows. Similarly, daily NCFs are stacked

to form a monthly stack. With an ensemble of monthly stacks, we conduct quality control by examining the consistency of NCFs, whereby correlation coefficients between all NCF pairs are computed and those with a below-average value are discarded. The accepted NCFs are stacked to obtain the final NCFs (i.e., $C^1$ functions), which form the basis for bridging asynchronous stations using the $C^2$ approach.

The $C^2$ workflow invokes source-receiver interferometry (SRI) to project the energy from one receiver via the surrounding backbone arrays to the other receiver[41]. The application of $C^2$ does not directly cross-correlate the noise recordings between the two target receivers, hence simultaneous operations of the two stations are not required. This idea can be applied to effectively tie asynchronous arrays (supplementary Figure S2). For two temporary arrays deployed at different time periods, we are able to retrieve the inter-array NCFs functions with the aid of the surrounding long-term stations via a three-step process. First, the $C^1$ is computed between temporary array A and the surrounding stations (supplementary Fig. S2a). Second, temporary array B, which is deployed after the extraction of temporary array A, is cross-correlated with the same set of stations. These two steps effectively turn the surrounding long-term stations into common virtual sources that illuminate both temporary arrays (supplementary Fig. S2b). Finally, for a target station pair, the two $C^1$ functions from the same virtual source are cross-correlated again to form a $C^2$ function, and all $C^2$ functions, each corresponding to a different virtual source, are stacked to obtain a final $C^2$ estimate. We perform a weighted stacking scheme based on the Voronoi cell tessellation and implement radial and azimuthal tapering as proposed in ref. [42] to improve the stacking. We refer readers to ref. [22] for implementation details. The $C^2$ workflow thus provides an indirect approach to retrieve the NCFs between asynchronous stations (or arrays), a situation that cannot be achieved with the conventional $C^1$ approach. The additional ray paths from $C^2$ connect asynchronous stations and provide complementary information to the $C^1$ dataset (supplementary Fig. S3a; also see Fig. 2). The waveforms of $C^1$ and $C^2$ both show clear surface wave energy with a similar move-out over a large (0–3500 km) distance range (supplementary Fig. S3b, c). We obtain a total of 230,788 and 696,046 NCFs from $C^1$ and $C^2$, respectively.

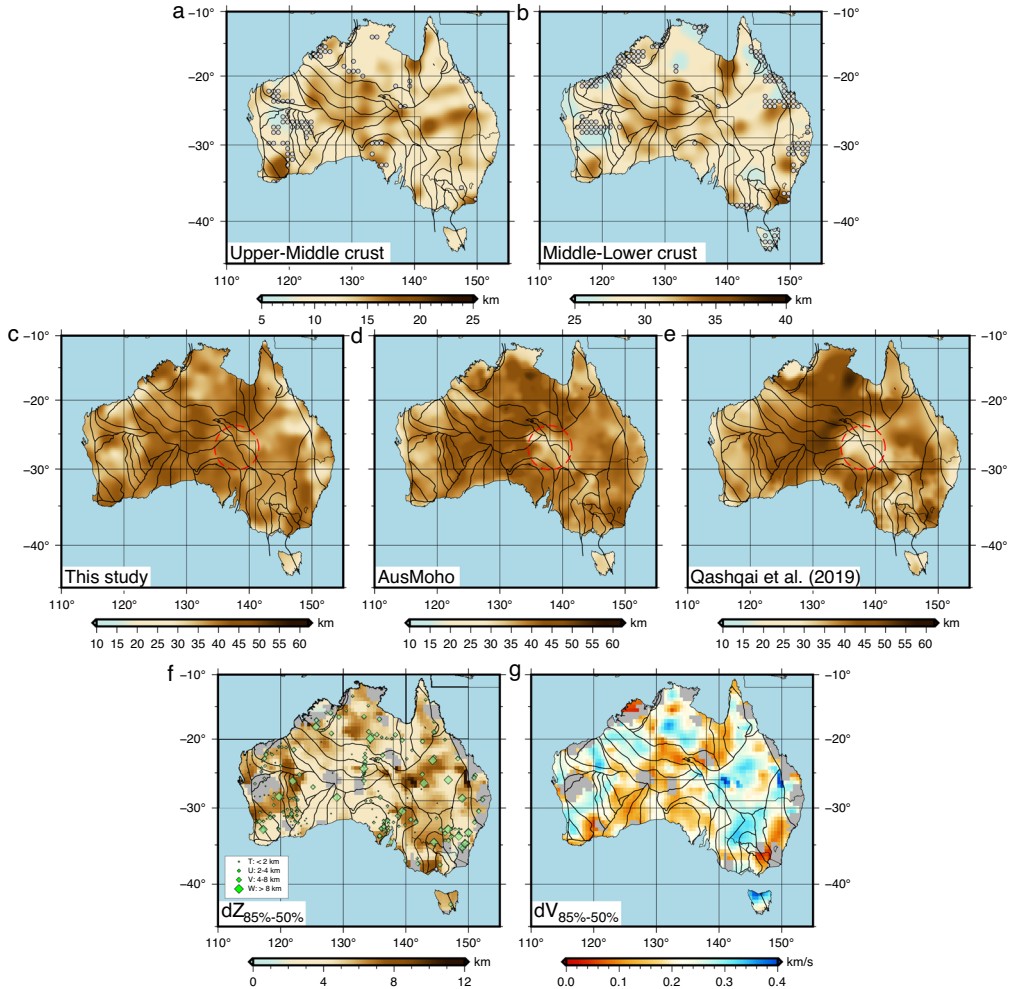

**Fig. 9 | Crustal layering determined from this study and previous work.** Crustal interfaces of **a** upper-middle crust and **b** middle-lower crust. The gray circles indicate the locations where the depth measurements are not reliable. Moho depths from **c** this study, **d** AusMoho[4], and **e** P-wave coda auto-correlation imaging[36]. The data used in ref. [36] is recorded by a subset of stations used in this study. The most significant Moho depth difference among these models is highlighted by the circle. **f** Transition thickness from the crust to the mantle. **g** The corresponding variation in shear velocity from the crust to the mantle. The transition thickness is obtained by taking the depth difference between the 50% and 85% increase from crust to mantle velocities. The gray areas mask the region where the Moho depth measurements are not reliable. The transition velocity is defined similarly by calculating the velocity difference. The green diamonds in **f** are the transition thickness determined from receiver functions from ref. [39].

## Dispersion measurements

We measure the surface wave group velocities of the NCFs from $C^1$ and $C^2$ using frequency-time analysis (FTAN)[24]. Several screening criteria are employed to select robust dispersion measurements including (1) a minimum signal-to-noise ratio (SNR) of 7, (2) a minimum inter-station distance that is three times greater than the wavelength[43], and (3) consistency between dispersion measurements from a similar distance range (supplementary Fig. S4). The last criterion applies a distance-dependent filter to the group velocities and eliminates measurements that deviate significantly (beyond one standard deviation) from the mean value of a distance bin. About 80% of the measurements are retained after quality control. The number of measurements is the largest between 5 and 10 seconds and gradually decreases towards longer periods (supplementary Fig. S5). The number of $C^2$ functions exceeds that of $C^1$ at periods <24 s, beyond which the quality of $C^2$ functions decreases sharply due to the relatively low SNR of the long-period signal. The mixture of signals recorded by broadband and short-period instruments in the higher-order cross-correlation calculation limits the bandwidth of the $C^2$ functions and hence long-period dispersion measurements (above 30 s) from $C^2$ are not used in the inversion. This issue can be potentially alleviated by optimizing the workflow by, for example, selecting only broadband stations as virtual sources and applying weighted stacking according to the frequency content of individual $C^2$ functions. These approaches will be investigated in a future study. To assess the quality of the measurements, we construct a cap-averaged group velocity map by placing the measurement values at the midpoints of the ray paths (supplementary Fig. S6), which approximates a simplified back-projection method of travel time inversion by only considering the contribution from the midpoints of the ray paths[44]. The resulting maps exhibit a large velocity variation across the Australian continent with well-defined velocity structures associated with major crustal blocks, which suggests the coherency of our group velocity measurements.

## Group velocity inversion

Group velocity is inverted using trans-dimensional Bayesian tomography[9]. This method employs an adaptive parameterization scheme and estimates uncertainties in model parameters, which is ideal for our dataset with a varying degree of data density (see Fig. 2). The choice of the noise level of data is important to constrain the smoothness of the model solution[9]. We determine this parameter by considering the resolving power of our data as indicated by the

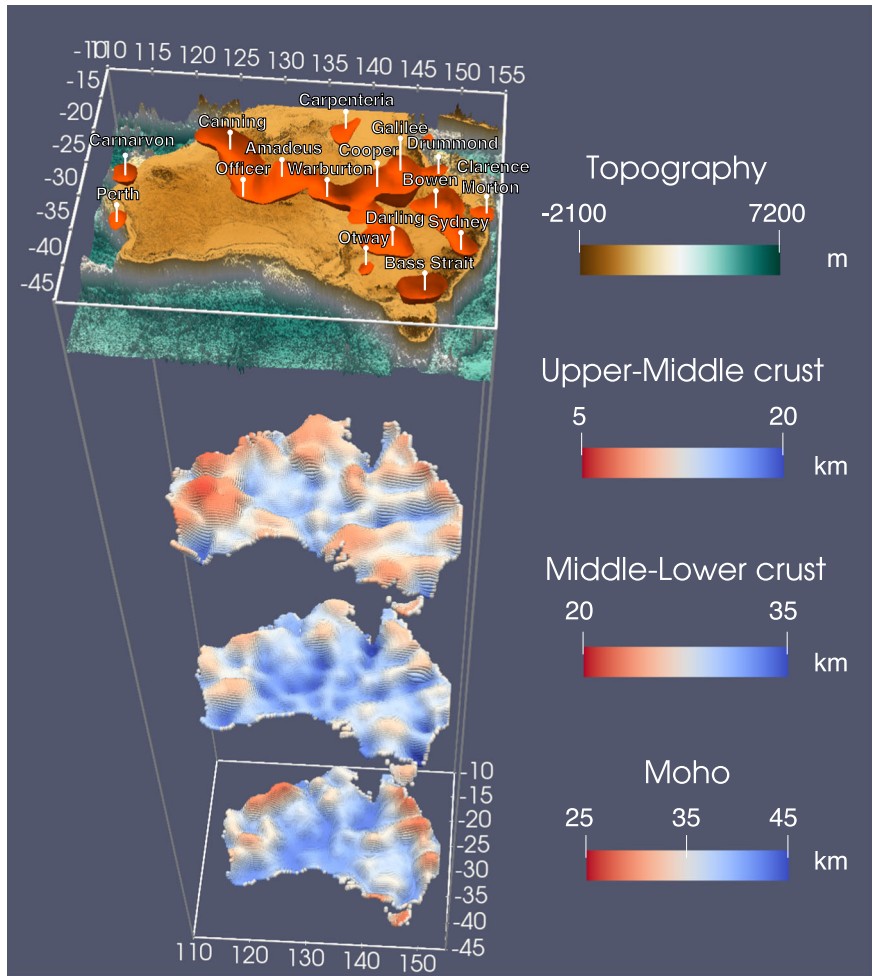

**Fig. 10 | A 3D rendering of the shear velocity model.** The sedimentary basins are represented by an isosurface of 3.1 km/s. The four interfaces are, from the top to bottom, surface topography, upper-middle crust boundary, middle-lower crust boundary, and the Moho.

checkerboard tests (see supplementary Fig. S7) such that the output velocity model mainly contains anomalies with sizes robustly constrained by the data. The input checkerboard pattern is best resolved in eastern Australia that is covered by dense seismic arrays. The level of recovery decreases towards northwestern Australia where velocity structures are smeared along the direction of surface wave propagation. The overall resolving power varies with the ray-path density from about 2-degree at periods <30 s (supplementary Fig. S7a, d) to about 4-degree at longer periods (supplementary Fig. S7e, f). The average model variance also increases with the period from ~0.2 km/s at 6 s to ~0.6 km/s at 46 s (supplementary Fig. S8). The spatial variation of variance is generally inversely related to the ray-path density.

## Shear velocity inversion

We invert a 1D shear velocity profile at each node location on a regularly spaced (0.75 deg) grid. We adopt a linear inversion algorithm from the code package of the Computer Programs in Seismology[45]. A well-known characteristic of linearized inversion is its strong dependence on the starting model. Fortunately, the first-order structure of the Australian continent is well constrained by earlier seismic studies. We construct an average 1D velocity profile from AuSREM[5] as the initial model and update the model iteratively while imposing smoothness constraints. To determine a proper damping parameter, we compute an average dispersion curve of all nodes and invert for an average velocity structure. The optimal damping value is determined by selecting the turning point of the trade-off curve between the data misfit and the L-2 norm of model parameters. A similar damping value is then applied to the inversion at all node locations. The change of damping values mostly affects the strength of the velocity perturbations and the trend of the velocity profile remains largely the same.

## Spatial relationship between shear velocities and mineral deposits

We compute the average velocity in the depth range of 0–5 km at each mineral deposit location, including a total of 3975 samples reported from ref. [46] (supplementary Fig. S9a). Then we summarize the velocities at all sample locations into a normalized cumulative density function (CDF), which indicates the occurrence frequency of certain velocity values at sampling locations. To test the statistical significance of the obtained CDF (i.e., the CDF for the observed spatial distribution of mineral deposits), we compute the CDF for a set of presumed deposit locations (supplementary Fig. S9b). In each trial, we assign each mineral deposit a random location within the continental land area and extract the velocity of its underlying crust. One simulation is completed after performing the same operation (i.e., location assignment and velocity extraction) for all mineral deposits. We conduct a total of 300 simulations for statistical analyses. The resulting CDFs are highly consistent and show a wider (2.8–3.7 km/s) distribution of seismic velocities than that from the real locations (2.9–3.7 km/s).

## Crustal interface measurements

We determine the major crustal interfaces using a velocity gradient approach. We respectively locate the two maxima of velocity gradient in the approximated depth range of the upper and lower crust, as well as the minimum in the middle crust. The gradient maxima mark the position of large variations in crustal property and correspond roughly to the depths of the basement and the Moho. We assume that the gradient minimum approximates the center of the middle crust that is characterized by relatively small velocity variations. Then the interface between the upper and middle crust is defined by the midpoint of the minimum and the upper maximum gradient. The interface between the middle and lower crust is defined correspondingly with the minimum and the lower maximum (supplementary Fig. S13). The two interfaces well delineate the transition region from high to low velocity gradients (supplementary Fig. S14). Our approach leads to reliable measurements at over 90% of grid points (see Fig. 9a, b). Nodes that are not well constrained are typically characterized by velocity profiles with small perturbations, hence smoothly varying velocity gradient without distinctive extrema. In addition, the Moho depth is quantitatively measured following the approach proposed by ref. [34]. We adopt the same criterion and use the depths where the velocity increases by 50% or 85% from the lower crust to the upper mantle as a proxy of the Moho. The cases of representative velocity profiles with thick, thin and undefined Moho transition are demonstrated in supplementary Fig. S15. Comparisons of our velocity profile with those obtained from receiver function inversions at nearby stations show that sharp velocity jump typically falls within the depth range determined from our model, and is closer to the shallow boundary (i.e., Z50; supplementary Figs. S16–S21). Hence, we adopt the shallow one as a proxy of the Moho depth. The difference between the two interfaces provides an estimate of the sharpness of the crust-mantle transition. We obtain reliable Moho depth measurements at ~90% of the inversion nodes. The undefined nodes are mainly caused by a lack of clear velocity gradient in the lower crust (see supplementary Fig. S15e, f), which are located near the continental margins where the data coverage is poor (see Fig. 10). The northern part of the Yilgarn craton in western Australia also exhibits a smooth crust-mantle transition, which prohibits the determination of reliable Moho transition thickness.

## Data availability

Broadband seismic waveforms are retrieved from IRIS-DMC (https://ds.iris.edu/ds/nodes/dmc/) and AusPass (http://www.auspass.edu.au). Velocity model obtained in this available from CSIRO data portable (https://data.csiro.au/collections/collection/CIcsiro:51008v1). The computer codes developed in this work are available upon request from corresponding authors.

## Code availability

The codes to compute the noise correlation functions are available from the corresponding author upon request.

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

## Acknowledgements

This work would not have been possible without decades of data collection efforts. We acknowledge AuScope, AusPass, Geoscience Australia, Australian National University, University of Melbourne, University of Western Australia and Geological Survey of Western Australia for collecting and sharing the seismic data. This research was fully funded by the Deep Earth Imaging Future Science Platform, CSIRO. This work is financially supported by the National Natural Science Foundation of China (42274059) to Y.C and Fundamental Research Funds for the Central Universities (NO. K20220232). This work was supported by resources provided by the Pawsey Supercomputing Centre with funding from the Australian Government and the Government of Western Australia.

## Author contributions

Y.C. processed the field data. Y.C., E.S., B.K., M.T., J.H., and D.L. analyzed the results. M.S provided geological background. E.S. designed the project. Y.C. wrote the manuscript. All authors revised the manuscript.

## Competing interests

The authors declare no competing interests.
