## [Peer Review File · Nature Communications]

Next-generation seismic model of the Australian crust from synchronous and asynchronous ambient noise imagingREVIEWER COMMENTS

Reviewer #2 (Remarks to the Author):

This manuscript presents a study of constructing a new seismic model of the Australian crust based on ambient noise tomography. Using 30 years of seismic recording from around 1600 stations, the authors constructed a crustal model with a resolution higher than models ever constructed for the Australian crust. Based on this 3D crustal model, they discussed various implications of their model. The two main focuses of their discussions are the implications of their model on mineral exploration and the evolution of the Australian lithosphere. By exploring the relationship between velocities at shallow depths and mineral locations, they show most known mineral locations in Australia are located at areas characterized by high velocities, that is the areas not covered by thick sedimentary layers. As to the discussions on the continental evolution, they claim their model has improved seismic constraints on the lower crust and uppermost mantle structures, which helps to better understand the evolution of the Australian continent as exemplified by the discussions on the nature of the Tasman line.

The manuscript is well written and the main points are clearly presented. However, I do not think the work is innovative or significant enough to warrant a publication in *Nature Communications*. I think it is more suitable to be published in a more specific journal such as the *Journal of Geophysics Research*. My arguments for the recommendation are detailed below.

This main contribution of this work is a new high-resolution 3D crustal model of Australia, which is enough for it to be publishable in journals like *JGR*. The authors claim their model could unlock resource potential in vast swaths of covered areas. However, I do not think this point is well supported. As it is well known, most mineral deposits across Australia are found in the areas with direct outcrops on the surface as thick sedimentary covers obscure the signature of mineral deposits, making potential mineral deposits hard to be discovered. So, it is not a surprise to see most known mineral deposits are founded in areas characterized by high velocities at the shallow depths.

As for the discussions on the upper mantle structures, when I examine the velocities of this model presented in Figure 9, the model of this study differs significantly from the other two models of AuSREM and CSEM-AU; while the latter two are very similar to each other. In addition, given the factor that there are only ~1300 paths of long-period data used in the tomography, I think the structures of the uppermost mantle in this study are not better constrained compared to the other two models. In addition, the discussions on the nature of the Tasman line seem not well supported by the model presented in Figure 9, which does not seem to clearly delineate a major velocity boundary across the Tasman line. I suggest the authors should focus their discussions more on the crustal structures.

Reviewer #3 (Remarks to the Author):

In this manuscript the authors developed an Australian wide shallow upper mantle V_s model by using an improved ambient noise tomography approach. Such a continental model was last developed nearly a decade ago. One significance here is the technical advance to apply cross-correlation of conventional noise correlation functions to achieve a relatively more uniform ray coverage for the continent. The resulting V_s model was then used to derive basin thickness and Moho topography across Australia, and then to infer the spatial correlation within known mineral deposits and to propose a tectonic rifting boundary across the Tasman line. The study bridges the gap across the

Moho depth between early crustal and shallow upper mantle velocity models, and therefore may contribute significantly to the next generation of the Australian reference model.

This reviewer raises two major points:

1 one major criticism is that several technical details seem needed to support the claims and current conclusions. Please refer to those listed in minor points below.

2 the 40-70 depths may not be best resolved by the ambient noise data set, and the authors included a lithospheric model from the full waveform inversion in their interpretation. More constraints may be considered to strengthen their point, for example, Pn, Sn and magnetotelluric models if available.

Minor points include

1 Line 46 – instead of quoting their own paper (ref 22), the authors may consider the original references here.

2 Section “Improvement of ray path coverage” - curious why the C2 coverage drops significantly after 24 s (or so). Presumably the permanent network stations are used as virtual sources so the signal to noise ratio should be good enough.

3 In the same section, about the longer period data – the authors claim in the last generation noise model, the data were only down to 30 s. In Figure 1b a lot more stations are used, but how many are broadband? Please provide ray coverage maps for longer periods.

4 The authors may consider providing depth sensitivity kernels in the suppl info.

5 The checkerboard resolution tests are fine for the periods considered, but unfortunately Figure 1 may suggest a strong station coverage difference across the Tasman line (please plot it on Figure 1). The authors may need to address if this would affect their conclusion of point 3 in the abstract, the tectonic rifting across the Tasman line.

6 It is disturbing to notice the large differences between the new model and the AuSREM model in both the absolute Vs (lines 112 to 120) and the Moho topography, e.g., the first 200 km along AA’; nearly 40km Moho across the Pilbara craton along BB’). In the latter case the authors may need to further justify the use of a universal 4.3 km/s Vs contour as the proxy for the Moho. Note this seems not consistent with the range of Vs contours described in line 174 (4.3-4.6 km/s)?

7 Point 6 above brings a question about the resolving power of the current dataset. In addition to plotting depth sensitivity kernels, the authors may consider inverting simple 1D synthetic velocity profiles to illustrate 1) the depth of sedimentary basins; and 2) the 40-70 km velocity contrasts across the Tasman line can be satisfactorily recovered.

8 Not clear the point Figure 7b and corresponding text try to make? At the first glance there does not seem to be deposits coincide with slow velocities (red).

9 Line 153 the choice of contour values (3.1 – 3.3. km/s) for the basement depth – why for the Moho is the Vs contour fixed?

10 Line 174 – please clarify if a fixed Vs contour is used? If not, how is the value adjusted between 4.3 and 4.6 km/s?

11 in the last section about the mantle structure – this is really not the prime region the current inversion can cover.

Reviewer #4 (Remarks to the Author):

Comments on “Next-generation seismic model of the Australian crust and implications for mineral resources and continental rifting” by Chen et al.

This paper presents a 3-D shear-wave velocity model of the Australian continent based on the latest results of ambient seismic noise tomography inversion. The authors adopted an innovative approach to increase data coverage by reconstructing the noise correlation functions between a pair of temporal stations deployed at different times but overlapped with a few permanent stations. This approach significantly improves the ray-path density that, in turn, leads to higher resolution of the tomography images. With the tomography results, the authors explain the highlights of the new model and discuss their implications for regional tectonics and mineral resources. Overall, this study is a significant step forward in our understanding of the detailed crustal and uppermost mantle velocity structures of the region. Before it is accepted for publication, however, I would like to point out a few issues that should be addressed and/or elaborated in more detail.

First of all, it is unclear how the authors define the ranges of upper, middle, and lower crust based on their results. My impression from Fig. 6 is that they roughly correspond to 0-10, 10-25, and 25-40 km, respectively. However, I think that the authors should be able to quantitatively define the 3-D geometry and thickness variation of the upper, middle, and lower crust for most part of the studied areas. The geometry and thickness variation can also provide important constraints on the characteristics of major geological structures (e.g., cratons, basins, and orogenic belts) and/or regional tectonic evolution.

Secondly, it is confusing when the authors discuss the spatial relationship between velocity anomalies and mineral deposits. Specifically, the authors point out in L145 that giant mineral deposits are preferentially located within 100 km of the craton edge (presumably along the edge of high-velocity anomalies), but later state in L149-150 that the deposits are found near basin margins (i.e., the edge of low-velocity anomalies). The text between L146 and L149 explains that basins can form along craton edge zones due to continental rifting. However, Fig. 1a clearly show that not all basins are located along craton edge zones. Fig. 7 also show that many mineral deposits (especially those in western Australia) are within the high-velocity craton. Thus, I am not sure if a generalized relationship can be derived.

Finally, the comparison of Moho depth with previous models is very useful. But there are cases where the extreme values should be treated with caution. For example, the new model has many places with the Moho depths larger than 55-60 km (Fig. 8). How confident can we trust these values? It is worth noting that some of these extreme values may be related to the way we define the Moho depth. In case that the velocity contrast between the lower crust and the uppermost mantle spans across a finite depth range (e.g., the Canadian Cordillera), the Moho "depth" (which is just one value, not a depth range) can differ by over 10 km depending on whether the 50%, 75%, or 100% of the velocity increase is chosen (similar examples can be found in Kao et al., 2013, JGR). I suggest the authors to quantitatively estimate the sharpness of the Moho discontinuity and its variation across Australia. The authors can also examine its spatial relationship with major tectonic/geological components and discuss the corresponding implications.

Some minor comments are given below for the authors' reference.

1. The authors use the word "wavespeed" in some places and "velocity" in other places. While technically "wavespeed" (a scalar) is the correct term, the seismological community has been using the term "velocity" to describe tomographic anomalies for decades. Whichever the authors prefer, they should be consistent throughout the text.
2. (L71, L216, and L288) The expression of "between A-B" is grammatically incorrect. It should be either "between A and B" or "in the range of A-B."
3. (L114) Citation to Figure 6d is probably erroneous. It should be Figure 6e.
4. (L275) The word "for" is missing before "its implementation details."
5. (L287) "is largest" should be "is the largest."

Honn Kao
Geological Survey of Canada

2022-09-14

Response to the reviews of manuscript

“Next-generation seismic model of the Australian crust and implications for mineral resources and continental rifting”

by Yunfeng Chen, Erdinc Saygin, Brian Kennett, Mehdi Torq Qashqai, Juerg Hauser, David Lumley and Mike Sandiford

We thank three reviewers for their constructive comments on our manuscript. We have made thorough corrections and addressed all these comments. This document provides our point-to-point responses to the questions and comments raised by the reviewers. Please also check the revised manuscript with marks for the detailed modifications. The original comments are in *Italics*, replies are in **Bold**, and call-outs to the revised manuscript are shown in **blue**. The line number in the responses are according to the modified edition.

Reviewer #2 (Remarks to the Author):

This manuscript presents a study of constructing a new seismic model of the Australian crust based on ambient noise tomography. Using 30 years of seismic recording from around 1600 stations, the authors constructed a crustal model with a resolution higher than models ever constructed for the Australian crust. Based on this 3D crustal model, they discussed various implications of their model. The two main focuses of their discussions are the implications of their model on mineral exploration and the evolution of the Australian lithosphere. By exploring the relationship between velocities at shallow depths and mineral locations, they show most known mineral locations in Australia are located at areas characterized by high velocities, that is the areas not covered by thick sedimentary layers. As to the discussions on the continental evolution, they claim their model has improved seismic constraints on the lower crust and uppermost mantle structures, which helps to better understand the evolution of the Australian continent as exemplified by the discussions on the nature of the Tasman line.

The manuscript is well written and the main points are clearly presented. However, I do not think the work is innovative or significant enough to warrant a publication in Nature Communications. I think it is more suitable to be published in a more specific journal such as the Journal of Geophysics Research. My arguments for the recommendation are detailed below.

This main contribution of this work is a new high-resolution 3D crustal model of Australia, which is enough for it to be publishable in journals like JGR. The authors claim their model could unlock resource potential in vast swaths of covered areas. However, I do not think this point is well supported. As it is well known, most mineral deposits across Australia are found in the areas with direct outcrops on the surface as thick sedimentary covers obscure the signature of mineral deposits, making potential mineral deposits hard to be discovered. So, it is not a surprise to see most known mineral deposits are founded in areas characterized by high velocities at the shallow depths.

Reply: We thank the reviewer for your comments. From a technical point of view, this study highlights the development of an innovative workflow for ambient noise imaging, which plays an important role in improving the resolution of the seismic model. Indeed, the new seismic model forms the backbone of our study and enables us to examine the crustal structure at unprecedented resolution. We believe that this work has significant and broad implications for other areas of the globe, rather than only simply presenting a new model to the Australian geophysical community. This is the reason why we submit the paper to Nature Communications for broader readership and impact in the geoscience community.

As far as mineral deposit locations, in this paper, we summarized two possible explanations for the distribution pattern of mineral deposits in Australia. One explanation is just as the reviewer suggested that the mineral deposits are buried underneath the thick cover and only those that are exposed or shallowly buried have been discovered. This implies a great potential for mineral exploration under the cover and also highlights the necessity of resolving detailed sedimentary structures with geophysical methods.

An alternate hypothesis is that the minerals are preferentially deposited. In Australia, a recent study has reported the correlation between the craton edge and giant mines (Hoggard et al., 2021) and found a significant portion (85%) of sediment-hosted base metals occur within 200 km of the transition between thick and thin lithosphere. This view links the mineral deposit formation to deep earth process. Taking advantage of our new model, we investigate the relationship between crustal structures and mineral deposits in more detail in this revised manuscript. We extracted velocities at mineral deposit locations and compared them to those at other locations (Figure R1). We find that the crustal velocities beneath mineral deposits are systematically faster. This is particularly evident at the upper (0-10 km) and lower (25-40 km) crustal depths. Such a relationship is reliable according to statistical tests. This observation suggests that the distribution of mineral deposits may not purely reflect the sampling bias due to shallow structures (e.g., the location where the outcrop exists) but possibly indicate a crustal-scale process. There could be a causal relationship between the deep (e.g., mantle upwelling) and shallow processes (mineral deposition). We have added this new analysis and expanded this part of the discussion in the revised manuscript (Lines 147-173).

“We further extract velocities in different depth intervals to examine if such a relationship persists to greater depths. To reduce the sampling bias caused by the clustering of deposits, the nearby deposits (within a 0.2-deg cell) are grouped to form a single sampling point. Our analysis shows that mean shear velocities are consistently faster beneath the mineral deposits than the model average (Figure 8). The mean velocities of the mineral deposit group are 0.06 km/s faster in the shallow (0-10 km) crust and 0.03 km/s faster in the deep (30-40 km) crust and are slightly faster (about 0.01 km/s) in the middle crust (10-30 km). The reliability of the difference in mean values of the two distributions is evaluated using the *t*-test, which assess the validity of the hypothesis that the mean of mineral deposit group is greater than the continental average. We obtain large *t* scores at all depths including the middle crust where the velocity difference is the small, showing a *t* score of 2.94 with a *p* value of 0.002 (see Figure 8). The test results indicate that the observation of consistently faster crustal velocities beneath the mineral deposits is statistically significant. This distinctive pattern suggests that the mineralization process likely involves the whole crust, not just the shallow portion. A corollary is that the distribution of mineral deposits may not solely reflect the sampling bias in mineral exploration due to the presence of sedimentary cover (e.g., the location where outcrop exists). There has been growing evidence that the formation of mineral deposits is closely related to deep magmatic processes controlled by lithospheric-scale structures (Griffin et al., 2013; Hoggard et al., 2020; Groves&Santosh, 2021). For instance, a recent study has reported a close spatial association between lithospheric gradient zones and sediment-hosted deposits around the globe (Hoggard et al., 2020). In Australia, this study showed that giant mines were preferentially located within 100 km of the craton edge that marks a transition in lithospheric thickness (Hoggard et al., 2020). One possible mechanism to form such a lithospheric boundary is through a continental rifting event in an extensional setting (Mckenzie 1978; Kuszniir &Ziegler, 1992), during which a basin subsides as a consequence of syn-rift mechanical stretching and post-rift isostatic re-equilibration (Kuszniir &Ziegler, 1992). However, not all basins and their associated sediment-hosted deposits are related to continental rifting on the craton margin. A notable exception is the Bowen basin in eastern Australia that has undergone contemporaneous thermal and foreland-loading subsidence in the Late Permian (Brakel et al., 2009). Nonetheless, the deep-seated structures such as the basin-bounding faults could facilitate the transport of geothermal fluids and thus play a critical role for the genesis and concentration of the sediment-hosted base metal deposits near the basin margins. Away from the basin margins, there are a significant portion of mineral deposits located in high velocity areas of the cratonic region in Western Australia. These deposits yet still form prominent clusters that align parallel to domain boundaries or elongate E-W along a high-velocity structure in central Yilgarn craton (see Figure 7a). We argue that these structural lineaments could mark weak/fracture zones that channel the mineralizing fluids and control the deposition sites.”

Finally, we acknowledge that much more effort is required to unlock the resources than just presented in this study. However, updated knowledge of crustal structures from a geophysical perspective marks an important step forward toward understanding the formation and deposition processes of mineral resources. We also believe that this work can also inspire future multi-disciplinary studies that involve mineralogy, geochemistry, geology, and geophysics for a more comprehensive understanding of the mineral systems in Australia.

Figure R1 The comparison of velocities beneath the mineral deposits (red) and other locations (blue) in different depth ranges. The histogram shows distribution of shear velocity and is normalized. The curves in corresponding colors are the Gaussian functions that best fit the histograms. The blue and red vertical lines indicate the mean velocity for each group. The statistic value from the t -test is indicated in the upper left corner.

As for the discussions on the upper mantle structures, when I examine the velocities of this model presented in Figure 9, the model of this study differs significantly from the other two models of AuSREM and CSEM-AU; while the latter two are very similar to each other. In addition, given the factor that there are only ~1300 paths of long-period data used in the tomography, I think the structures of the uppermost mantle in this study are not better constrained compared to the other two models. In addition, the discussions on the nature of the Tasman line seem not well supported by the model presented in Figure 9, which does not seem to clearly delineate a major velocity boundary across the Tasman line. I suggest the authors should focus their discussions more on the crustal structures.

Reply: We thank the reviewer for raising this concern about model resolution. We agree with the reviewer that the upper mantle is not the best-constrained depth range in our model. To examine the robustness of the velocity structures, we conducted a resolution test using hypothetical structures that contain a sharp velocity contrast across the Tasman line (Figure R2). The recovered model using the same inversion parameters as the real data show that the input structures can be well recovered. This suggests that the ray-path coverage is sufficient to constrain the first-order velocity structure at longer periods.

We also agree with the reviewer that the Tasman is not characterized as a major velocity boundary in our model. In fact, the most intriguing observation is the reversal of velocity patterns across the Tasman line

from shallow to deep depths. The observed large-scale velocity variation in both lateral and vertical directions is robustly constrained by our data and inversion method based on our thorough resolution analyses (see our reply to Reviewer 4 for details). Therefore, we propose a continental rifting model that can provide a possible explanation for the observed seismic structures, but this does not rule out other interpretations of the Tasman line. A well-developed discussion on the nature of the Tasman line requires more effort than presented in this work. Therefore, to make this paper more concise and focused, we follow the suggestions from Reviewers2&3 and mainly present the crustal structure and remove most of the discussions on the Tasman line.

Figure R2 (a) Input and (b) output models of the hypothesis test. The inset in (a) shows the ray-path coverage at 45 s. The inset in (b) shows the variance of seismic velocity from the trans-dimensional inversion.

We thank Reviewer 2 for the constructive comments that have helped to improve this manuscript.

Reviewer #3 (Remarks to the Author):

In this manuscript the authors developed an Australian wide shallow upper mantle V_s model by using an improved ambient noise tomography approach. Such a continental model was last developed nearly a decade ago. One significance here is the technical advance to apply cross-correlation of conventional noise correlation functions to achieve a relatively more uniform ray coverage for the continent. The resulting V_s model was then used to derive basin thickness and Moho topography across Australia, and then to infer the spatial correlation within known mineral deposits and to propose a tectonic rifting boundary across the Tasman line. The study bridges the gap across the Moho depth between early crustal and shallow upper mantle velocity models, and therefore may contribute significantly to the next generation of the Australian reference model.

Reply: We thank the reviewer for the positive comments. It is our hope that this study will provide a good example of how this advance in seismology can contribute to an improved understanding of the Earth's structure.

This reviewer raises two major points:

1 one major criticism is that several technical details seem needed to support the claims and current conclusions. Please refer to those listed in minor points below.

Reply: We provide more technical details regarding ray-path coverage, resolution analysis, interface measurement and sensitivity test in the revised manuscript. Please see our replies to the points below for further details.

2 the 40-70 depths may not be best resolved by the ambient noise data set, and the authors included a lithospheric model from the full waveform inversion in their interpretation. More constraints may be considered to strengthen their point, for example, P_n , S_n and magnetotelluric models if available.

Reply: Thank you for this suggestion. To further validate our model, we extracted the average velocity of the uppermost mantle (50-70 km) from a recent S_n velocity model (Wei et al., 2018). Figure R3 shows the comparison of the three models. They all show a similar first-order structural variation with low velocities observed beneath the center and the eastern edge of the continent. This comparison indicates that our model does have sensitivity to velocity structures in this depth range. However, the exact shape and strength of velocity anomalies are more difficult to constrain and compare because of the difference in data type/coverage, imaging technique, and model resolution between various studies. This figure has been added to the revised manuscript (Figure 11).

Figure R3 Average shear velocity in the depth range of 50-70 km from (a) this study, (b) AuSREM (Kennett et al., 2013) and (c) S_n velocity model (Wei et al., 2018).

Minor points include

1 Line 46 – instead of quoting their own paper (ref 22), the authors may consider the original references here.

Reply: The C^2 workflow used in this study was first proposed in our previous work (Chen and Saygin, 2020). We have cited the paper by Zhang et al. (2020) that proposed a similar idea as our study.

2 Section “Improvement of ray path coverage” - curious why the C^2 coverage drops significantly after 24 s (or so). Presumably the permanent network stations are used as virtual sources so the signal to noise ratio should be good enough.

Reply: We thank the reviewer for raising this important question. This is mainly caused by the procedure of C^2 calculation, which correlates (again) the correlation functions from different frequency bands. In particular, the short-period stations that are deployed mainly in eastern Australia count for a significant portion of the data used in our study (Figure R4). These stations typically have a corner frequency of around 10 s, hence the longer-period information cannot be preserved. Excluding the short-period stations may help improve the SNR at longer periods. This, however, also reduces the number of stations used in C^2 , which in turn limits the SNR. Improvement to SNR can be potentially made by designing a workflow that can distinguish the long- and short-period stations and apply a weighted stacking scheme (e.g., based on frequency content). This could help preserve a broad frequency bandwidth while ensuring sufficient stacks during the C^2 calculation. We have clarified this point in Lines 317-321 of the revised manuscript.

“The mixture of signals recorded by broadband and short-period instruments in the higher-order cross-correlation calculation limits the bandwidth of the C^2 functions and hence long-period dispersion measurements (above 30 s) from C^2 are not used in the inversion. This issue can be potentially alleviated by optimizing the workflow by, for example, selecting only broadband stations as virtual sources and applying weighted stacking according to the frequency content of individual C^2 functions. These approaches will be investigated in a future study.”

3 In the same section, about the longer period data – the authors claim in the last generation noise model, the data were only down to 30 s. In Figure 1b a lot more stations are used, but how many are broadband? Please provide ray coverage maps for longer periods.

Reply: Figure R4 shows the broadband and short-period stations, which are obtained from the AusPass website. Most of the transportable arrays in Australia are short-period stations, whereas the recently deployed permanent stations (e.g., S1 and AU networks) are primarily broadband stations. We have included the ray-path coverage in the group velocity results (Figure 3 in the revised manuscript).

Figure R4 The distribution of broadband (left) and short-period (right) stations in Australia. Note that some of these stations have recently become open datasets.

4 The authors may consider providing depth sensitivity kernels in the suppl info.

Reply: We have computed the sensitivity kernel (Figure R5) and included this plot in the supplementary information (Figure S10). The long-period (e.g., 45 s) surface waves can provide sensitivity to structures in the upper mantle (e.g., 50 km depth).

Figure R5 (a) Velocity model obtained by averaging the AuSREM model and is used to compute (b) the group velocity sensitivity kernel.

5 The checkerboard resolution tests are fine for the periods considered, but unfortunately Figure 1 may suggest a strong station coverage difference across the Tasman line (please plot it on Figure 1). The authors

may need to address if this would affect their conclusion of point 3 in the abstract, the tectonic rifting across the Tasman line.

Reply: We thank the reviewer for raising this concern. We have designed a hypothesis test to examine if the unbalanced station coverage can affect the resolvability of the uppermost mantle structures. We introduce bands of low and high-velocity structures with widths of 4 degrees to either side of the Tasman line (TL) (Figure R2a). We consider the station distribution at the long period of 45 s, at which the ray path coverage is high across the TL. The inversion can robustly recover the structural variation well and shows the minimum uncertainty across the TL (Figure R2b). This test suggests that the large-scale structures are well constrained at longer periods and the observations near the TL are valid.

In this study, we only provide a possible interpretation for the nature of TL that can be consistent with our seismic observations reasonably well. A more detailed interpretation of TL requires evaluating other mechanisms in the context of new seismic observations, which is beyond the scope of the current study. To make this paper more concise and the theme more focused, we follow the suggestions from Reviewers 2&3 and mainly present the crustal portion of the model. We now only report the robust observations in the uppermost mantle and remove the discussions of the TL.

6 It is disturbing to notice the large differences between the new model and the AuSREM model in both the absolute Vs (lines 112 to 120) and the Moho topography, e.g., the first 200 km along AA'; nearly 40km Moho across the Pilbara craton along BB'). In the latter case the authors may need to further justify the use of a universal 4.3 km/s Vs contour as the proxy for the Moho. Note this seems not consistent with the range of Vs contours described in line 174 (4.3-4.6 km/s)?

Reply: We thank the reviewer for pointing this out. The difference in absolute velocity between our model and AuSREM can be attributed to two main factors: 1) The shear velocity of AuSREM is mainly constructed from receiver function (RF) inversion and ambient noise tomography. While the RF inversion is sensitive to the crustal interface, it is less reliable in terms of absolute velocity. 2) The previous ambient noise model was constructed with about 10% of the ray paths used in this study. The long-period information is particularly sparse in the previous model. Hence, the dispersion data is mainly used to constrain the upper 20 km in AuSREM, and the shear velocities from the ambient noise model below 30 km are not incorporated in AuSREM.

In this revision, we found that the initial model overestimated the upper mantle velocities due to the linear extrapolation of lower crust velocity. We have reconstructed the initial model and run the inversion. The updated model leads to slightly different velocity structures at lower crust/uppermost mantle depths. The results of the velocity comparison are shown in Figure R6 (also see Figure 6 in the revised manuscript).

Figure R6 Comparison of upper, middle and lower crustal structures between (a)-(c) our model and (d)-(f) AuSRREM (Salmon et al., 2013). (g)-(i) The corresponding difference between the two models. The black dashed lines in (a) and (d) indicates velocity contours of 3.45 km/s

We apologize for the confusing definition of the velocity contour. We did not use a constant velocity contour to determine the Moho depth in our study. Instead, we used varying velocity values to consider the lateral variation in average velocity across the continent. In this revision, we follow the suggestion from Reviewer 4 and use a more objective criterion that invokes velocity jump to determine the Moho. The Moho is determined by the depth of 50% velocity jump from the lower crust to upper mantle velocities. This leads to a different Moho depth map compared to that obtained previously (Figure R12; also see Figure 9 in the revised manuscript).

7 Point 6 above brings a question about the resolving power of the current dataset. In addition to plotting depth sensitivity kernels, the authors may consider inverting simple 1D synthetic velocity profiles to illustrate 1) the depth of sedimentary basins; and 2) the 40-70 km velocity contrasts across the Tasman line can be satisfactorily recovered.

Reply: We thank the reviewer for these suggestions. We have designed two synthetic tests to examine the resolving power of our dataset and provided these tests in the supplementary material. In the first test, we vary the sediment thicknesses from 2 to 10 km (Figure R7), which represents a typical depth range

observed in our study. We then generate a synthetic dispersion curve for each model and invert for shear velocities using the same parameters as those used in the real data cases. The test results show that the low-velocity sedimentary basin structures can be well recovered, indicating that the shallow structures are well constrained by our short-period data.

Figure R7 Inversion of 1D velocity structure with varying sediment thickness.

In the second test, we systematically vary the velocity from -10% to 10% in the upper mantle depth (40-70 km) (Figure R8). The inversion successfully recovered the trend of velocity but failed to resolve the discontinuities. This is expected from the inversion of the surface wave dispersion curve which is sensitive to average velocity over a certain depth range (depending on the frequency) rather than sharp boundaries. This test suggests our data is sufficient to distinguish the first-order shear velocity contrast in the uppermost mantle.

Figure R8 Inversion of 1D velocity structure with varying velocity perturbations in upper mantle.

8 Not clear the point Figure 7b and corresponding text try to make? At the first glance there does not seem to be deposits coincide with slow velocities (red).

Reply: We used Figure 7b to quantitatively assess the relationship between crustal velocity and mineral deposit location. There are only a small portion (26%) of the mineral deposits that fall within or, more precisely speaking, hit the margin of the low-velocity zone. As reviewer suggested, the vast majority of mineral deposits are underlain by relatively high crustal velocities.

We have added a new figure (Figure R1; also see Figure 8 in the revised manuscript) to better demonstrate this relationship and support our point. We find that the crustal velocities are systematically faster at the mineral deposit locations. This observation suggests that the mineral deposition process may involve the whole crust not only the shallow structures. Please see our reply to the question from Reviewer 2 for more details.

9 Line 153 the choice of contour values (3.1 – 3.3. km/s) for the basement depth – why for the Moho is the V_s contour fixed?

Reply: We apologize for the confusion. The Moho depth is in fact determined using a range of velocities not a fixed value in order to consider the lateral variation in average crustal velocity.

10 Line 174 – please clarify if a fixed V_s contour is used? If not, how is the value adjusted between 4.3 and 4.6 km/s?

Reply: Based on the suggestion from Reviewer 4, we now determine the Moho depth using the criterion of velocity jump. Please see our reply to Reviewer 4 for details of the method.

11 in the last section about the mantle structure – this is really not the prime region the current inversion can cover.

Reply: We agree with the reviewer. This concern was also raised by Reviewer 2. To address this point, we have 1) conducted resolution analysis and only reported robust observations in this depth range, 2) removed the discussions on the nature of the Tasman line and its implication on continental rifting of the Australian continent, and 3) conducted more quantitative analysis of our model and expanded the discussions on the crustal structures per the suggestions from Reviewer 4.

We thank Reviewer 3 for the constructive comments that have helped to improve this manuscript.

Reviewer #4 (Remarks to the Author):

Comments on “Next-generation seismic model of the Australian crust and implications for mineral resources and continental rifting” by Chen et al.

This paper presents a 3-D shear-wave velocity model of the Australian continent based on the latest results of ambient seismic noise tomography inversion. The authors adopted an innovative approach to increase data coverage by reconstructing the noise correlation functions between a pair of temporal stations deployed at different times but overlapped with a few permanent stations. This approach significantly improves the ray-path density that, in turn, leads to higher resolution of the tomography images. With the tomography results, the authors explain the highlights of the new model and discuss their implications for regional tectonics and mineral resources. Overall, this study is a significant step forward in our understanding of the detailed crustal and uppermost mantle velocity structures of the region. Before it is accepted for publication, however, I would like to point out a few issues that should be addressed and/or elaborated in more detail.

Reply: We thank the reviewer for acknowledging the contribution of our study. We have addressed the concerns raised. Please see below for details.

First of all, it is unclear how the authors define the ranges of upper, middle, and lower crust based on their results. My impression from Fig. 6 is that they roughly correspond to 0-10, 10-25, and 25-40 km, respectively. However, I think that the authors should be able to quantitatively define the 3-D geometry and thickness variation of the upper, middle, and lower crust for most part of the studied areas. The geometry and thickness variation can also provide important constraints on the characteristics of major geological structures (e.g., cratons, basins, and orogenic belts) and/or regional tectonic evolution.

Reply: We thank the reviewer for this comment. The reviewer is correct that we divided the model into different layers according to their presumed depth ranges. This is mainly for the purpose of ease of comparison for different models. We have followed the reviewer’s suggestion and conducted quantitative analysis of our model to determine the 3D geometry of crustal interfaces. For 1D velocity profile at each inversion grid point, we locate the depths of maximum velocity gradient in upper and lower crust (Figure R9). Because the middle crust is generally characterized by less heterogeneous structures, we define the center of the middle crust as the depth with the lowest velocity gradient (local minimum). Then the crustal boundaries are determined by the midpoint of the local maximum and minimum. This criterion leads to robust measurements at most of the grid points. The resulting crustal interfaces are shown in Figure R10. The depth variation of crustal interfaces is generally similar to that of the Moho. The crustal interfaces are elevated in the cratonic region of western and northern Australia and are depressed in central Australia where the Moho is also the deepest. We have added these analyses and discussions on Lines 193-198 the revised manuscript.

“We determine, for the first time, major crustal layering of the Australian continent using a velocity gradient approach (see Method section). The shallow crustal interface that approximately divides the upper and middle crust is on average 10 km deep, and is generally deeper beneath the sedimentary basins and shallower in the cratonic regions in western, northern and southern Australia (Figure 9a). The lower interface separating the middle and lower crust resides at about 27 km depth, which generally mimics the shallower layer with the most significant depression observed in central and southeastern Australia (Figure 9b).”

Figure R9 An example measurement of crustal interfaces. (a) A 1D shear velocity profile. The black circles indicate the depths where the velocity gradient reaches local extrema. The red and blue crosses indicate the depth of the upper-middle and middle-lower crustal boundaries, respectively. (b) The velocity gradient of shear velocity profile shown in (a). The depths of gradient extrema and crustal boundaries are indicated by the dashed lines in corresponding colors.

Figure R10 Crustal layering determined from this study and previous work. Crustal interfaces of (a) upper-middle crust and (b) middle-lower crust.

Secondly, it is confusing when the authors discuss the spatial relationship between velocity anomalies and mineral deposits. Specifically, the authors point out in L145 that giant mineral deposits are preferentially located within 100 km of the craton edge (presumably along the edge of high-velocity anomalies), but later state in L149-150 that the deposits are found near basin margins (i.e., the edge of low-velocity anomalies). The text between L146 and L149 explains that basins can form along craton edge zones due to continental rifting. However, Fig. 1a clearly show that not all basins are located along craton edge zones. Fig. 7 also show that many mineral deposits (especially those in western Australia) are within the high-velocity craton. Thus, I am not sure if a generalized relationship can be derived.

Reply: We thank the reviewer for this comment. We apologize for the confusion. The arguments on L145 (high velocity) and L149-150 refer to observations at different depths. The argument on L145 was quoted from a recent study (Hoggard et al., 2020), which reported that the giant mineral deposits were preferentially located near the craton edges (see figure below). The location of high-velocity craton edge is mapped at upper mantle depths from shear velocity SL2013sv (Schaeffer and Lebedev, 2013), not our model. The argument on L149-150 refers to low-velocity sedimentary basin from our model.

The text from L146-149 indeed associates the basin formation with continental rifting. The rift basin formed close to the craton margin can explain some giant mineral deposits like those reported in Hoggard et al. (2020). However, with respect to all deposits in Australia, these arguments are inaccurate considering variable mechanisms of basin subsidence across the continent. We intend to say that there is likely a structural control on the mineral deposition process. There has been increasing agreement that the spatial distribution of mineral deposits is closely related to lithospheric-scale structure. The basin margin, where large bounding faults could exist, provides a potential pathway for the transfer of mineralizing fluids. We emphasize that the presence of large-scale faults is not necessarily related to basin margin. As pointed out by the reviewer, the mineral deposits in western Australia are not associated with any low-velocity structures. Rather they are located either along the domain boundaries or along linear high-velocity structures found in this study. These structural lineaments could provide weak/fracture zones that channel the mineralizing fluids. As suggested by the reviewer, we cannot derive a general relationship between different sizes/types of mineral deposits and structural types in this study. Our purpose is to examine if there is a systematic trend in seismic velocity distribution at mineral deposit locations, which could help us improve the understanding of the potential first-order structural control on mineralization. We have clarified this point and rewritten the discussions on Lines 144-173 of the revised manuscript.

Finally, the comparison of Moho depth with previous models is very useful. But there are cases where the extreme values should be treated with caution. For example, the new model has many places with the Moho depths larger than 55-60 km (Fig. 8). How confident can we trust these values? It is worth noting that some of these extreme values may be related to the way we define the Moho depth. In case that the velocity contrast between the lower crust and the uppermost mantle spans across a finite depth range (e.g., the Canadian Cordillera), the Moho “depth” (which is just one value, not a depth range) can differ by over 10 km depending on whether the 50%, 75%, or 100% of the velocity increase is chosen (similar examples can be found in Kao et al., 2013, JGR). I suggest the authors to quantitatively estimate the sharpness of the Moho discontinuity and its variation across Australia. The authors can also examine its spatial relationship with major tectonic/geological components and discuss the corresponding implications.

Reply: We thank the reviewer for these comments. The extreme values in Moho depths are measurement outliers caused by the instability of our previous depth determination approach. We have followed the approach proposed by Kao et al. (2013) and determined the depth of Moho using the velocity jump. Figure R11 shows an example of depth measurement. Similar to Kao et al. (2013), we used a 50% or 85% increase from lower crust to upper mantle velocities as a proxy of Moho depths. We found that an 85% velocity increase overestimates the Moho depth as compared to existing constraints, and a 50% velocity increase provides a good estimate of the Moho (Figure R12).

Indeed, the crust-mantle transition can span a certain distance range depending on the nature of Moho (sharp vs. gradient). We have considered this issue by systematically mapping the thickness of crust-mantle transition. To this end, we follow the approach proposed by Kao et al. (2013) and define the transition thickness as the depth difference between the 50% and 85% velocity increase. In addition, we compare the resulting thickness map with the Moho sharpness constrained by receiver functions (Kennett and Saygin, 2015). Both studies show that there is a considerable variation in transition thickness at both continental (thousands of km) and regional (tens to hundreds of km) scales (Figure R13). Similar observations include the sharp Moho near craton margins and gradient Moho in eastern Australia. However, we do not observe a clear relationship between Moho sharpness and the age of crustal blocks. This may suggest a significant amount of reworking of continental crust that changes the composition and rheology of the lower crust and uppermost mantle. This new analysis has been added to Lines 229-245 of the revised manuscript.

“The sensitivity of dispersion data to deep structure (supplementary Figures S10 and S12) enables us to characterize the transition in physical properties from crust to mantle. We measure the transition thickness (i.e., Moho sharpness) and its corresponding velocity variation by considering the depth and velocity difference between the 50% and 85% of velocity increases from crust to mantle (Kao et al., 2013). The Moho sharpness (Figure 10a) is generally anti-correlated with the velocity jump (Figure 10b), wherein a smaller velocity increase leads to a sharper boundary and vice versa. The Moho sharpness map shows large (> 6 km) transition thicknesses in western, northern and eastern Australia, whereas zones of relatively thin (< 4 km) crust-mantle transition dominate central Australia and extend southward to the coastline (see Figure 10a). Our measurements from ambient noise imaging are compared with the constraints from receiver functions that classify the crust-mantle transition into four distinctive groups (Kennett & Saygin, 2015). Receiver function imaging reveals large variation in transition thickness across the continent and at a regional scale of hundreds of kilometers. Similar observations between the two studies include 1) sharp (2-4 km) Moho along the northern and southeastern edges of the Yilgarn craton and considerable variability in the cratonic interior of Western Australia, 2) sharp Moho in southern Australia, particularly in the vicinity of the Gawler craton, and 3) thick transition regions beneath the sedimentary basins (e.g., Eramanga and Cooper basins) in eastern Australia. Overall, our observations do not show a clear relationship of Moho sharpness to tectonic age. For example, a thick crust-mantle transition is observed beneath both Archean and Phanerozoic basements. This could suggest that the rheological properties near the base of crust are not only inherited from crustal formation but may have undergone substantial reworking during the secular evolution of the continental crust (Kennett & Saygin, 2015).”

Overall, benefiting from the reviewer’s suggestions, we are able to construct a 3D model of crustal geometry (Figure R14). This model summarizes the key contributions of our study and provides a more comprehensive view of the crustal structures of the Australian continent.

Figure R11 An example measurement of Moho depth. (a) A 1D shear velocity profile. The blue and black circles mark the depths of lower crust and upper mantle velocities. The red crosses mark 50% and 85% velocity jump from lower crust to upper mantle. (b) The velocity gradient profile of shear velocity in (a). The dashed lines in corresponding colors mark the depths defined in (a).

Figure R12 Moho depths from (a) this study, (b) AusMoho (Kennett et al., 2011) and (c) P-wave coda auto-correlation imaging (Qashqai et al., 2019). The data used in Qashqai et al. (2019) is recorded by a subset of stations used in this study. The most significant Moho depth difference among these models is highlighted by the circle.

Figure R13 The transition in physical properties from crust to mantle of the Australian continent. (a) Transition thickness. (b) Transition velocity. The transition thickness is obtained by taking the depth difference between the 50% and 85% increase from crust to mantle velocities. The transition velocity is defined similarly by calculating the velocity difference. The green diamonds in (a) are the transition thickness determined from receiver functions from Kennett&Saygin (2015).

Figure R14 A 3D rendering of the velocity model. The sedimentary basins are represented by the isosurface of 3.1 km/s. The four interfaces are, from the top to bottom, surface topography, upper-middle crust interface, middle-lower crust interface and the Moho.

Some minor comments are given below for the authors' reference.

1. The authors use the word "wavespeed" in some places and "velocity" in other places. While technically "wavespeed" (a scalar) is the correct term, the seismological community has been using the term "velocity" to describe tomographic anomalies for decades. Whichever the authors prefer, they should be consistent throughout the text.

Reply: We thank the reviewer for this comment. We have used "velocity" consistently in the revised manuscript.

2. (L71, L216, and L288) The expression of "between A–B" is grammatically incorrect. It should be either "between A and B" or "in the range of A–B."

Reply: Corrected as suggested.

3. (L114) Citation to Figure 6d is probably erroneous. It should be Figure 6e.

Reply: Corrected as suggested.

4. (L275) The word "for" is missing before "its implementation details."

Reply: Corrected as suggested.

5. (L287) "is largest" should be "is the largest."

Reply: Corrected as suggested.

Honn Kao

Geological Survey of Canada

We thank Reviewer 4 for the insightful comments that have helped to improve this manuscript.

REVIEWER COMMENTS

Reviewer #3 (Remarks to the Author):

Glad to see my comment regarding Fig. 7 in the previous submission is now expanded to two figures. The new Fig. 8 is suggestive of the whole-crust control of the velocity structures to the mineral systems, but can a conclusion be made that there is more spatial correlation between the deposits and lower crust high velocities? This might have immediate implications for the magma processes, e.g., lower crustal underplating.

Reviewer #4 (Remarks to the Author):

I appreciate the authors' visible effort to address my previous comments. I am satisfied with the revisions, especially about the newly added discussions on the depth variation of crustal interfaces and the transition from lower crust to uppermost mantle. I have a few questions specifically about the new results and would like the authors to elaborate in more detail.

The authors state that "Our approach leads to reliable measurements at the majority of grid points (see Figures 9a and b)" in L366-367. I wonder if the authors can provide a more quantitative description by giving the percentage instead of using the vague word "majority." Also, can the authors elaborate why the proposed approach cannot work for some of the grid points? It would be helpful if the authors show some representative examples of the inverted Vs models at grid points that are characterized by a sharp Moho or a thick crust-mantle transition, and a few examples of those "abnormal" cases.

More importantly, where are these "abnormal" Vs grid points located geographically? Are they simply randomly scattered across the entire continent (I guess not!), or do they appear to cluster in specific regions? Another important question is the robustness of these abnormal Vs models. Are they mainly artifacts due to imperfect inversion or poor data resolution? A direct comparison of the Vs models derived by this study and that by previous studies with receiver function inversion for some of the representative grid points would probably help us assess the quality and robustness of individual Vs models.

A few minor comments are listed below for the authors' reference.

(L88) It might be more straightforward to simply specify "At the shallow depth of 1 km, ..." rather than "At shallow (1 km) depth, ..."

(L137) "of Yilgarn craton" should be "of the Yilgarn craton."

(Figure 1) The authors mention "Cooper basin" several times in the text. It is marked in Figures 4, 7, and 12, but not in Figure 1a. I don't think that the authors should be blamed for this negligence because Figure 1a is adopted directly from Raymond (2018). Nonetheless, it might be worth the effort to mark the approximate location of the Cooper basin in Figure 1a to make it consistent with the rest of the paper. The authors can cite appropriate reference(s) in the figure caption as needed.

2022-12-11

Response to the reviews of manuscript

“Next-generation seismic model of the Australian crust and implications for mineral resources and continental rifting”

by Yunfeng Chen, Erdinc Saygin, Brian Kennett, Mehdi Torq Qashqai, Juerg Hauser, David Lumley and Mike Sandiford

We thank two reviewers for their constructive comments on our manuscript. We have addressed their remaining concerns. This document provides our point-to-point responses to the questions and comments raised by the reviewers. Please also check the revised manuscript with marks for the detailed modifications. The original comments are in *Italics*, replies are in **Bold**, and call-outs to the revised manuscript are shown in blue. The line number in the responses are according to the modified edition.

Reviewer #3 (Remarks to the Author):

Glad to see my comment regarding Fig. 7 in the previous submission is now expended to two figures. The new Fig. 8 is suggestive of the whole-crust control of the velocity structures to the mineral systems, but can a conclusion be made that there is more spatial correlation between the deposits and lower crust high velocities? This might have immediate implications for the magma processes, e.g., lower crustal underplating.

Reply: We thank the reviewer for pointing out this interesting interpretation. Our model indeed supports a potential crustal-scale process for mineral deposition. However, based on our spatial correlation analysis alone we cannot clearly differentiate the contribution of crustal structures at different depth levels to the spatial distribution pattern of mineral deposits. For example, the distribution of mineral deposits agrees nicely with the high velocities at shallow (0-5 km) depths in western Australia (see Figure 7a), whereas in eastern Australia along the coastline, the mineral deposits show better correlation with deep structures (see the velocity map at 32 km in Figure 4e). We think that the dominating controlling factor is likely variable across the continent. More detailed seismic imaging with dense arrays and examination of regional lower crust structures are required to understand the potential role of lower crustal process (e.g., underplating) in mineral deposition. We have clarified this point in the revised manuscript (Line 174-177).

We thank Reviewer 4 for constructive comments on our model implications on mineral systems that improve the impact of this study.

Reviewer #4 (Remarks to the Author):

I appreciate the authors' visible effort to address my previous comments. I am satisfied with the revisions, especially about the newly added discussions on the depth variation of crustal interfaces and the transition from lower crust to uppermost mantle. I have a few questions specifically about the new results and would like the authors to elaborate in more detail.

The authors state that “Our approach leads to reliable measurements at the majority of grid points (see Figures 9a and b)” in L366-367. I wonder if the authors can provide a more quantitative description by giving the percentage instead of using the vague word “majority.” Also, can the authors elaborate why the proposed approach cannot work for some of the grid points? It would be helpful if the authors show some representative examples of the inverted Vs models at grid points that are characterized by a sharp Moho or a thick crust-mantle transition, and a few examples of those “abnormal” cases.

More importantly, where are these “abnormal” Vs grid points located geographically? Are they simply randomly scattered across the entire continent (I guest not!), or do they appear to cluster in specific regions? Another important question is the robustness of these abnormal Vs models. Are they mainly artifacts due to imperfect inversion or poor data resolution? A direct comparison of the Vs models derived by this study and that by previous studies with receiver function inversion for some of the representative grid points would probably help us assess the quality and robustness of individual Vs models.

Reply: We thank the reviewer for the comment on the velocity model. We have carefully examined our measurements of both crustal interfaces and the Moho. For the intra-crustal discontinuities, we have slightly modified the selection criteria by reducing the threshold of minimum velocity gradient that are used to identify the peaks (local maxima). This helps to improve the interface picking in south Australia near the Gawler craton where the velocity gradient perturbation is relatively small. As a result, we obtained reliable measurements at over 90% of the inversion nodes (a total of 2520), with 2268 and 2416 good measurements for the shallow and deep crustal interfaces, respectively (Figure R1). For the nodes that are not well constrained, the underlying crustal structures are typically characterized by velocity with small perturbations, hence smoothly varying velocity gradient without distinctive extrema. These nodes are generally located in cratonic regions of western Australia as well offshore regions that are not well covered by ray paths (Figure R2). We have clarified this point in the revised manuscript in line 360-363. The plot of cross-section of gradient measurements is provided in the supplementary material.

“The two interfaces well delineate the transition region from high to low velocity gradients (supplementary Figure S14). Our approach leads to reliable measurements at over 90% of grid points (see Figures 9a and b). Nodes that are not well constrained are typically characterized by velocity profiles with small perturbations, hence smoothly varying velocity gradient without distinctive extrema.”

Figure R1. Crustal interfaces of (a) upper-middle crust and (b) middle-lower crust. The gray circles indicate the locations where the depth measurements are not reliable.

Figure R2. The cross-sections of our model showing vertical velocity gradients. The locations are shown in Figure R1. The gray circles represent major crustal interfaces. The Moho extracted from our model is indicated by the purple dashed line.

For the Moho depth, we demonstrate the cases of representative velocity profiles with thick, thin and undefined Moho transition (Figure R3). We obtained reliable Moho depth measurements at about 90% of the inversion nodes. The undefined nodes are mainly caused by a lack of clear velocity gradient in the lower crust, with the majority of them located near the continental margins where the data coverage is poor. The northern part of the Yilgarn craton in western Australia also exhibits a smooth crust-mantle transition (Figures R3e-f), which prohibits the determination of reliable Moho transition thickness. We have updated the figure and added these discussions in the Method section of the revised manuscript (line 365-374).

“The cases of representative velocity profiles with thick, thin and undefined Moho transition are demonstrated in supplementary Figure S15. Comparisons of our velocity profile with those obtained from receiver function inversions at nearby stations show that sharp velocity jump typically falls within the depth range determined from our model, and is closer to the shallow boundary (i.e., Z50; supplementary Figures S16-S21). Hence, we adopt the shallow one as a proxy of the Moho depth. The difference between the two interfaces provides an estimate of the sharpness of the crust-mantle transition. We obtain reliable Moho depth measurements at about 90% of the inversion nodes. The undefined nodes are mainly caused by a lack of clear velocity gradient in the lower crust (see supplementary Figures S15e-f), which are located near the continental margins where the data coverage is poor (see Figure 10). The northern part of the Yilgarn craton in western Australia also exhibits a smooth crust-mantle transition, which prohibits the determination of reliable Moho transition thickness.”

Figure R3. Representative velocity profiles with (a) thick, (c) thin and (e) undefined Moho transition thickness. The blue and black circles mark the depths of lower crust and upper mantle velocities. The red crosses mark 50% and 85% velocity jump from lower crust to upper mantle. (b, d, f) The corresponding velocity gradient profile of shear velocity. The dashed lines in corresponding colors mark the depths determined from shear velocities.

Figure R4. The transition in physical properties from crust to mantle of the Australian continent. (a) Transition thickness. (b) Transition velocity. The transition thickness is obtained by taking the depth difference between the 50% and 85% increase from crust to mantle velocities. The gray areas mask the region where the Moho depth measurements are not reliable. The transition velocity is defined similarly by calculating the velocity difference. The green diamonds in (a) are the transition thickness determined from receiver functions from Kennett and Saygin (2015). The blue stars mark the locations of representative velocity profiles demonstrating the characteristics of Moho transition (Figure R3). The red stars mark the stations where shear velocities from receiver function inversions are compared with our velocity model (Figures R5-R8).

Finally, we have compared the velocity profiles with earlier constraints from receiver function inversions (RFs). We digitized the shear velocity profiles reported from literature at a few targeting locations across the continent. First of all, the velocity profiles obtained from ambient noise tomography (ANT) are much smoother compared to those from receiver function inversions. Specifically, ANT is insensitive to sharp

discontinuities that can be often recovered from RFs. At the same time, we notice that the RF inversion results in Australia can exhibit large variabilities among nearby stations. For example, crustal velocities at stations WT09 and WT10 deployed in eastern Yilgarn craton (Western Australia) are respectively higher and lower than that of our model (Figure R5). Our velocity profile typically falls within the range of velocities determined from nearby stations (e.g., stations WT03, WT04, WT05 in western Yilgarn craton). This suggests that the velocity model from ANT represents a regional average structure instead of a point-based measurement as from RF inversion. The difference between ANT and RF inversion results can also be a consequence of different sensitivity of the two methods. Surface wave dispersion curve is more sensitive to absolute velocities of the crust whereas RF is mainly sensitive to sharp discontinuities (i.e., velocity gradient). This is particularly evident from the station BL10 in central Australia (Figure R8), where we observe a large difference between the two velocity profiles. We find that our velocity profiles are generally in good agreement with those from AuSREM that constrain absolute crustal velocities by integrating a variety of observations.

The Moho depth from RFs typically falls within the depth range determined by the 50% and 85% velocity jumps, and is closer to the shallow boundary (i.e., Z50; Figures R5 and R7). However, we also found large differences at stations deployed in the Pilbara craton (Figure R6), where depth measured from ANT is considerably deeper (by 5-10 km). This could suggest that the long-period dispersion information is not well retrieved in this region. This also highlights the necessity of jointly inverting multiple observations (e.g., receiver functions, surface wave dispersion curves, auto-correlation functions) to reconcile the resolution and sensitivity of different methods and develop a more accurate model in the future. We have added these plots to supplementary material.

Figure R5. Left panel: (a) Shear velocity profiles beneath stations WT09 and WT10 deployed in eastern Yilgarn craton. (b) Shear velocity gradient from our model. Right panel: (a) Shear velocity profiles beneath stations WT03, WT04, WT05 three deployed in western Yilgarn craton. The velocity profiles are reported from Reading et al. (2003).

Figure R6. Similar to Figure R5 but for two stations deployed in Pilbara craton. The velocity profiles are reported from Reading and Kennett (2003).

Figure R7. Similar to Figure R5 but for two stations deployed in southeastern Australia. The velocity profiles are reported from Bello et al. (2021) and Fontaine et al. (2013).

Figure R8. Similar to Figure R5 but for station BL10 deployed in central Australia. The velocity profile is reported from Sippl (2016).

A few minor comments are listed below for the authors' reference.

(L88) It might be more straightforward to simply specify “At the shallow depth of 1 km, ...” rather than “At shallow (1 km) depth, ...”

Reply: Corrected as suggested.

(L137) “of Yilgarn craton” should be “of the Yilgarn craton.”

Reply: Corrected as suggested.

(Figure 1) The authors mention “Cooper basin” several times in the text. It is marked in Figures 4, 7, and 12, but not in Figure 1a. I don't think that the authors should be blamed for this negligence because Figure 1a is adopted directly from Raymond (2018). Nonetheless, it might be worth the effort to mark the approximate location of the Cooper basin in Figure 1a to make it consistent with the rest of the paper. The authors can cite appropriate reference(s) in the figure caption as needed.

Reply: Thank you for this suggestion. We have updated Figure 1a and outlined the Copper basin.

We are grateful for the two rounds of constructive comments from Reviewer 4 that significantly improve this study.

REVIEWERS' COMMENTS

Reviewer #4 (Remarks to the Author):

I thank the authors for their excellent effort in addressing my comments. I am generally very satisfied with their responses. I enjoy reading this paper.

I notice that the authors took out one figure (Figure 11 in the previous version) and its discussion in this revised version. Since no explanation is given for this removal, I assume that the authors made such a decision to limit the total length of this paper (or maybe for other reasons as well). In contrast, I think that the authors probably have added a bit too much discussion about future studies. For example, I would suggest the authors to delete the sentence "We believe that the integrated efforts..." (L263-265) as it looks redundant after reading those sentences immediately above (L256-263).

In any case, I don't think that this paper will need another round of review. Whether the authors accept my cosmetic suggestion or not will not affect the value of this paper. I congratulate the authors for making a very important contribution to the tomography of crustal structures in Australia.

2023-01-23

Response to the reviews of manuscript

“Next-generation seismic model of the Australian crust and implications for mineral resources and continental rifting”

by Yunfeng Chen, Erdinc Saygin, Brian Kennett, Mehdi Tork Qashqai, Juerg Hauser, David Lumley and Mike Sandiford

We thank the reviewer for their constructive comments on our manuscript. We have addressed their remaining concerns. This document provides our point-to-point responses to the questions and comments raised by the reviewer. Please also check the revised manuscript with marks for the detailed modifications. The original comments are in *Italics*, replies are in **Bold**.

Reviewer #4 (Remarks to the Author):

I thank the authors for their excellent effort in addressing my comments. I am generally very satisfied with their responses. I enjoy reading this paper.

I notice that the authors took out one figure (Figure 11 in the previous version) and its discussion in this revised version. Since no explanation is given for this removal, I assume that the authors made such a decision to limit the total length of this paper (or maybe for other reasons as well). In contrast, I think that the authors probably have added a bit too much discussion about future studies. For example, I would suggest the authors to delete the sentence “We believe that the integrated efforts...” (L263-265) as it looks redundant after reading those sentences immediately above (L256-263).

In any case, I don't think that this paper will need another round of review. Whether the authors accept my cosmetic suggestion or not will not affect the value of this paper. I congratulate the authors for making a very important contribution to the tomography of crustal structures in Australia.

Reply: We thank the reviewer for the kind words and acknowledgment of our study. We removed Figure 11 in the last round of revision for the considerations of 1) reducing the length of the paper as only a maximum of 10 figures is allowed, and 2) making the theme of the paper more focused by just presenting the crustal structures. While our resolution analysis shows that the large-scale structures in the uppermost mantle are reliably resolved, the model resolution is still far from ideal compared with the crustal counterpart, and future efforts are required to further refine the model in this depth range. We also agree that the discussion on future work in the summary paragraph is redundant and have removed it accordingly.

We are grateful for the three rounds of constructive comments from Reviewer 4 that significantly improve this study.